# Probabilistic Tracking of Annual Cropland Changes over Large, Complex Agricultural Landscapes Using Google Earth Engine

**Sitian Xiong** [1,†], **Priscilla Baltezar** [1,2,*,†], **Morgan A. Crowley** [3], **Michael Cecil** [1], **Stefano C. Crema** [4], **Eli Baldwin** [1], **Jeffrey A. Cardille** [3,5] **and Lyndon Estes** [1]

1   Graduate School of Geography, Clark University, Worcester, MA 01610, USA
2   Department of Geography, College of the Social Sciences, University of California, Los Angeles, CA 90095, USA
3   Department of Natural Resource Sciences, McGill University, Sainte-Anne-de-Bellevue, QC H9X 3V9, Canada
4   Clark Labs, Clark University, Worcester, MA 01610, USA
5   Bieler School of Environment, McGill University, Montréal, QC H3A 2A7, Canada
*   Correspondence: pbaltezar@g.ucla.edu
†   These authors contributed equally to this article.

**Abstract:** Cropland expansion is expected to increase across sub-Saharan African (SSA) countries in the next thirty years to meet growing food needs across the continent. These land transformations will have cascading social and ecological impacts that can be monitored using novel Earth observation techniques that produce datasets complementary to national cropland surveys. In this study, we present a flexible Bayesian data synthesis workflow on Google Earth Engine (GEE) that can be used to fuse optical and synthetic aperture radar data and demonstrate its ability to track agricultural change at national scales. We adapted the previously developed Bayesian Updating of Land Cover (Unsupervised) algorithm (BULC-U) by integrating a shapelet and slope thresholding algorithm to identify the locations and dates of cropland expansion and implemented a tiling scheme to allow the processing of large volumes of imagery. We apply this approach to map annual cropland change from 2000 to 2015 for Zambia (750,000 km$^2$), a country that is experiencing rapid growth in agricultural land. We applied our cropland mapping approach to a time series of unsupervised classifications developed from Landsat 5, 7, 8, Sentinel-1, and ALOS PALSAR within 1476 tiles covering Zambia. The annual cropland changes maps reveal active cropland expansion between 2000 to 2015 in Zambia, especially in the Southern, Central, and Eastern provinces. Our accuracy assessment estimates that we have identified 27.5% to 69.6% of the total cropland expansion from 2000 to 2015 in Zambia (commission errors between 6.1% to 37.6%), depending on the slope threshold. Our results demonstrate the usefulness of Bayesian data fusion and shapelet, slope-based thresholding to synthesize optical and synthetic aperture radar for monitoring agricultural changes in situations where training data are scarce. In addition, the annual cropland maps provide one of the first spatially continuous, annually incremented accounts of cropland growth in this region. Our flexible, cloud-based workflow using GEE enables multi-sensor, national-scale agricultural change monitoring at low cost for users.

**Keywords:** agriculture; change detection; Bayesian fusion; Zambia; Google Earth Engine

## 1. Introduction

Africa's food needs are expected to more than triple during the next 30 years, and at least 140 million hectares of new cropland will be needed to satisfy this demand [1–3]. Such a large-scale land transformation, which is already underway [4,5], will have substantial social, economic, political, and environmental consequences that will be felt from local to global scales [2,6,7].

Given the rapid pace of change and their potential consequences, it is essential to measure and monitor agricultural changes as they unfold over the continent, across the range of spatial scales that they will impact. The demand and production for food supplies

are jeopardized by the challenges of climate change, disease, pests, and increasingly lower productivity and yields urging for more effective and timely monitoring [8]. Conducting such monitoring is difficult, as comprehensive national survey datasets are typically lacking, out of date, and of low spatial resolution [4,9,10]. As such, the only feasible way to undertake this task is with remote sensing. However, developing accurate, remotely sensed maps of Africa's smallholder-dominated agricultural systems is hard to accomplish [11–14]. For one, the nature of these systems—small fields, often mixed in with the surrounding vegetation types [13]—is a major source of confusion for the models used to convert remote sensing imagery into maps, while the spatial resolution of the imagery is often too coarse to separate small-scale agriculture from the surrounding vegetations [13,15–19]. In addition to this, the data needed to train and assess mapping models is either lacking or not publicly available, and is expensive to develop [12,13,16,17,20,21], while mapping over large regions at frequent time steps is a major computational challenge [13] with higher spatial resolution imagery. Given these limitations, the available regional land cover maps that include agricultural land tend to be infrequently produced (e.g., made once or twice per decade) and may contain substantial errors [9,20–22]. Moreover, the available maps tend to be produced for different years and from separate efforts, using different training data, models, and classification schemes, resulting in inconsistencies that make their combined use and comparison difficult [15,22]. These factors cause large uncertainties as to where and how much cropland exists [20,23] and how it is changing.

Fortunately, recent advances in Earth Observation methods are making it increasingly possible to conduct accurate, large-area monitoring over hard-to-map regions [16,24]. Among the first of these advances was the opening of the multi-decadal Landsat archive under the free and open data policy in 2008 [17,25], as well as the rapid growth in Earth-observing satellites, which are increasing the frequency, resolution, and depth of information available from spaceborne observations [18], while lowering the cost. This increasing number of observations enables continuous, frequent mapping of land cover, while providing detailed imagery that can be used to create the training and reference data (labels) required by mapping algorithms [9,19,26]. Synthetic aperture radar (SAR) sensors, such as Sentinel-1 and ALOS, can characterize agricultural landscapes [27] due to their spectral ability to actively illuminate a target regardless of atmospheric effects using microwave frequencies [28]. SAR has proved useful for estimating crop planting area, yield, and other crop variables [29], not only because of imperviousness to atmospheric effects, but also its sensitivity to the geometric structure and dielectric properties of crops [27,30,31]. Some studies have used multi-scale and multi-sensor approaches to understand field scale phenology by combining SAR and optical sensors resulting in a higher accuracy for mapping crop cover and yield [30,32]. In one study comparing the ability of Landsat, Sentinel-1, Sentinel-2, and the Moderate Resolution Imaging Spectroradiometer (MODIS) to analyze rapid changes in field scale phenology of corn and soy [30], high accuracies were achieved using individual sensors, with marginal improvements when combining all sensors. However, classification performance was poor in regions of persistent cloud cover, thus it was concluded that Sentinel-1 data may be critical for enabling crop mapping over large areas to mitigate the loss of information due to atmospheric effects [30]. Similarly, other studies have also used a sensor combination approach, but with radar sensors of different frequencies. In a study that evaluated the potential of SAR for early season corn, soy, and hay-pasture detection, both the TerraSAR-X and RADARSAT-2 sensors were compared [33]. These studies demonstrated the importance of using multiple sensors, albeit in the same wavelengths, to increase the opportunities for multiple observations and re-look opportunities.

There have also been large gains in the capabilities of the algorithms used to extract meaning from satellite imagery, e.g., [34], as well as image analytical approaches that help to minimize the requirements for training datasets. Such approaches include data fusion methods, which gather evidence from multiple datasets to create new hybrid products, such as cropland maps developed by synthesizing available land use land cover (LULC) products [35,36]. These approaches vary from those that rely on expert ranking of the qual-

ity of different input datasets [37] to probabilistic techniques such as the Bayesian Updating of Land Cover (BULC) algorithm [38–44], which weighs the evidence and uncertainty in different input maps while constructing LULC time series.

Another key development is the growth of cloud-based Earth Observation analytical platforms, most prominently GEE [24], which bring together the pre-processed image archives, algorithms, and computing power to enable their application at large scales. Such platforms have given rise to several important breakthroughs in monitoring, including the creation of 30 m resolution datasets for tracking annual changes in forest cover [45], monthly changes in surface water across the entire globe [46], urban change [47], as well as multi-decadal changes in irrigated cropland dynamics across major crop production regions [44].

Despite these tremendous gains, there are two primary challenges that need to be resolved to realize the capacity to remotely track the dynamics of complex smallholder-dominated croplands. First, there is a critical lack of labeled data for training and assessing the accuracy of mapping algorithms [16]. However, most algorithms that enable continuous tracking of land cover change [45] typically require many land cover labels, especially when the desired outcome includes classification of the type of change ("from to") [19]. The second challenge is that it is difficult to develop large area, long-term, high frequency (yearly to sub-yearly) datasets of moderate (10–30 m) resolution imagery over areas of persistent cloud cover [13,45,48,49].

In this study, we developed a mapping approach that addresses these outstanding challenges, to track annual agricultural change at a national extent in locations where training data are scarce and cloud cover is frequent. We demonstrate this approach by mapping cropland expansion in Zambia, a large country (~750,000 km$^2$) with an extensive and geographically diverse agricultural sector that is undergoing rapid agricultural change [5,50]. Cropland in Zambia is difficult to map accurately due to the predominance of small-scale agriculture and the frequent cloud cover during the growing season [10,23]. To overcome the challenge of limited training data, we used the Bayesian Updating of Land Cover (Unsupervised) algorithm (BULC-U), an automated version of BULC that was developed to detect change in long time series of Landsat imagery by combining unsupervised and object-based classification within the Earth Engine platform [37,38]. A key feature of BULC-U is that it greatly reduces the need for labeled data by using existing land cover maps to train the algorithm. In this study, we used a single existing 30 m land cover map to train BULC-U and extended BULC-U's capabilities by applying linear regression and shapelets [51] to estimate the year of agricultural expansion. To overcome the lower density of imagery due to high cloud cover while leveraging the capabilities of SAR for improving cropland classifications [31], we constructed a detailed satellite record using Landsat 5, 7, and 8 archives with SAR imagery collected by both the Sentinel-1 and the ALOS sensors. We implemented this approach on GEE [24], which provided the infrastructure in terms of access to pre-processed imagery, storage, and computational power [24] that we needed to conduct this large-scale analysis. Our results indicate the potential of this approach for fine-grained identification and analysis of agricultural expansion over large, hard-to-map areas.

## 2. Materials and Methods

### 2.1. Study Area

Zambia was chosen for our analysis because it is representative of other agriculturally rich countries in sub-Saharan Africa (SSA) that have experienced rapid crop expansion [52,53]. Zambia is mainly a sub-tropical climate characterized by three seasons: hot and dry (mid-August to mid-November), wet and rainy (mid-November to April), and a cool and dry season (May to mid-August) [52]. Zambia is a large country with an area of 752,612 km$^2$ that requires 41 Landsat scenes to cover its full extent (Figure 1). Zambia is categorized into three major agro-ecological regions (i.e., Regions I, II, and III) according to the countries' agronomist on precipitation, soils, and climatic attributes [52]. Across these agro-ecological regions, agriculture is primarily rain-fed with little commercial irrigation [53]. Small-scale

agriculture (farms < 5 Ha) is the most common form in Zambia, constituting more than three-quarters of existing farms, with the remaining share consisting of medium- (5–100 ha) and large-scale (>100 ha) farms [50]. Farm size distributions are changing rapidly, however, as domestic agricultural investment has driven a rapid increase in the number of medium-scale farms, which more than doubled between 2008–2014, and now occupy over 50% of Zambia's agricultural land [50]. These dynamics indicate substantial diversity in Zambia's agricultural systems, including substantial variations in the size and shape of fields and the crops they grow. Maize, groundnuts, soybeans, cassava, and cotton were the five most widely grown crops between 2015–2020 [54], with maize constituting over 50% of harvested area on average. Regions I (semi-arid area) and III (high rain-fall area) are predominantly small-scale farming systems that include the southern, eastern, western, and high northern areas of Zambia. Some of these regions' cropping systems include hand hoe, low input, shifting, and semi-permanent cultivation techniques [52]. Region II, on the other hand, has many commercial farms, which can be found in much of central Zambia [55]. The major crop types are diverse and include a combination of staple and commodity cultivation, in addition to dairy and livestock [52].

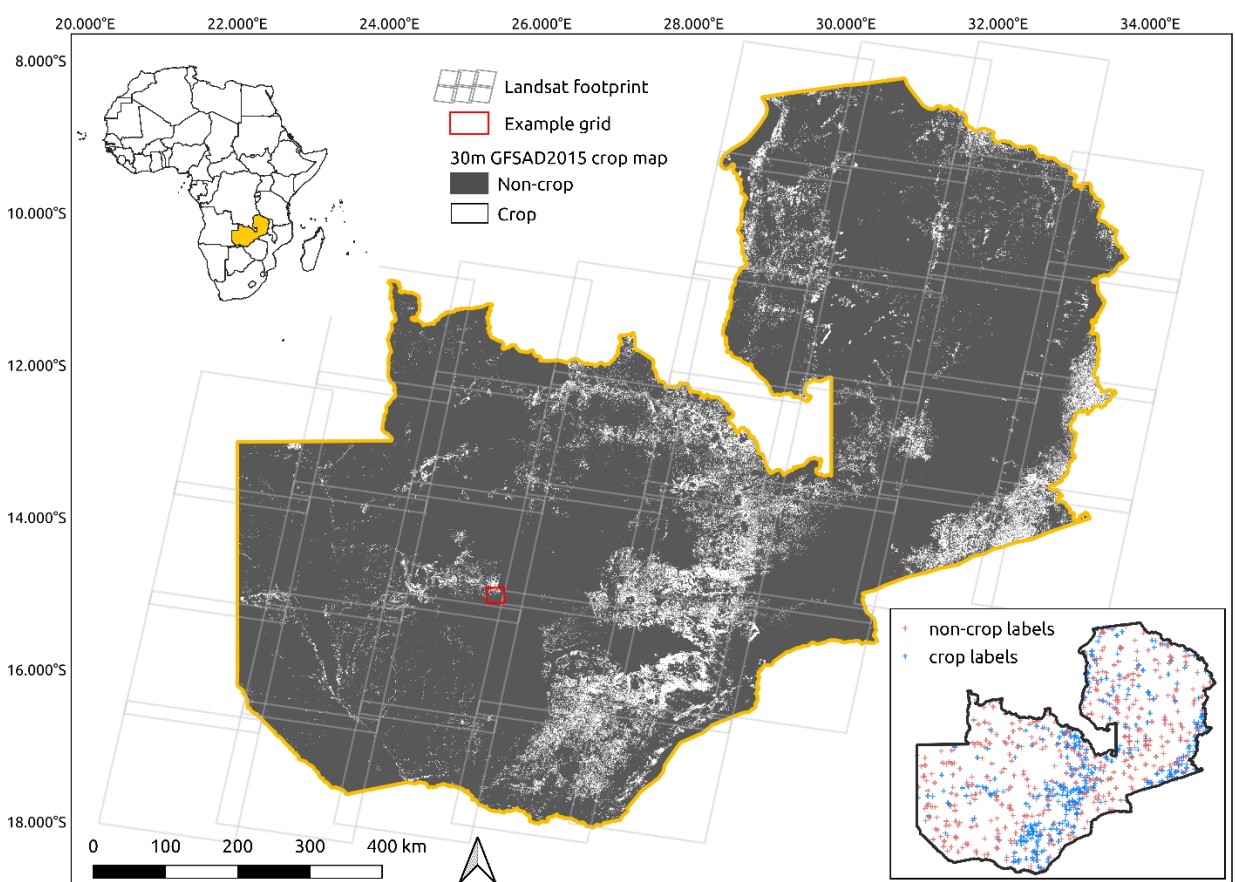

**Figure 1.** Study area figure showing Zambia study region, cropland coverage, and Landsat tile footprint. The 812 validation samples are shown in the lower right inset map, with cropland labels for 2015 shown in blue and non-cropland labels in red.

In the following section, we outline the variable inputs (see Table 1) and methods used in this study to track cropland changes in Zambia from 2000 to 2015 using GEE, following the workflow illustrated in Figure 2. Our processing workflow begins with the compilation of input data, including satellite imagery and the Global Food Security-Support Analysis Data (GFSAD) derived from the 2015 satellite imagery [56], the source for our binary crop base map starting in 2015. Satellite data was initially divided into smaller analytical units, and then classified using an unsupervised K-means approach. Resulting

unsupervised cropland maps were then updated by GFSAD backwards in time using the BULC-U algorithm. Finally, areas of cropland expansion were assessed over the 15-year time series to determine where and when cropland change occurred.

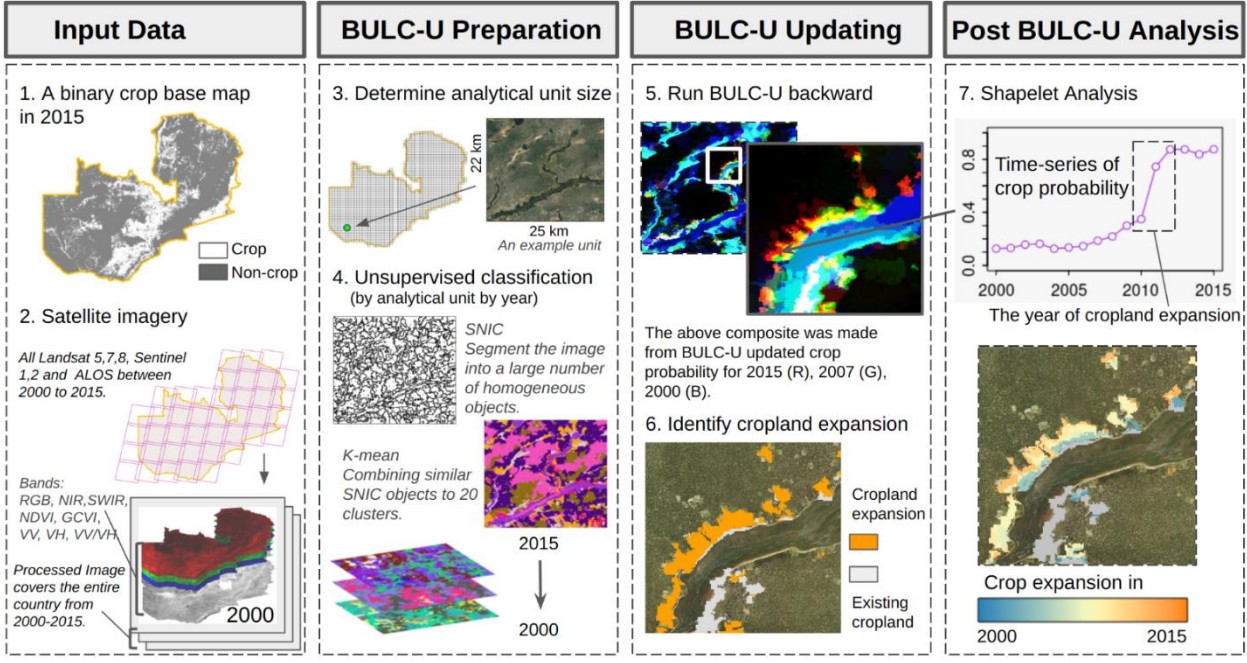

**Figure 2.** An overview of our research workflow, which begins with the preparation of input datasets. (**1**) Crop class extracted as mask from GFSAD. (**2**) Yearly optical and radar imagery spatially filtered and pre-processed. (**3**) Identification of minimum mapping unit. (**4**) Cluster identification using a K–means classifier and segmentation. (**5**) BULC–U Algorithm executed in descending order from 2015. (**6**) Determination of yearly cropland probabilities and expansion given the slope threshold (**7**) Determination of statistically significant shapelets of crop expansion over time.

## 2.2. Description of Datasets & Pre-Processing Methods

### 2.2.1. Basemap

The Bayesian fusion algorithm (BULC-U) requires an input reference dataset to inform the initial probabilities. We used the 30 m Global Food Security-Support Analysis Data (GFSAD) cropland raster dataset for the year 2015 as our reference map [56]. Through a combination of pixel-based and object-oriented classifiers in addition to a Recursive Hierarchical Image Segmentation approach, the NASA Making Earth System Data Records for Use in Research Environments (MEaSUREs) program was able to retrieve cropland extent at a 30-m spatial resolution. Each GFSAD GeoTIFF file contains a binary classification of cropland area and water bodies over a 10-degree by 10-degree area for the continent of Africa. Preliminary evaluation of the GFSAD product in Zambia using stratified sampling (see Section 2.3.4) had an overall accuracy of 81%, a producer's accuracy of 89%, and a user's accuracy of 73% for the cropland class. We clipped the GFSAD map using Zambia's national boundary and remapped the class values to their respective binary classes (cropland or non-cropland).

### 2.2.2. Landsat Annual Medoid Composites

To develop annual image composites, we utilized all available Collection 1 Level-1 Surface Reflectance scenes from the Landsat 8 Operational Land Imager (L8 OLI) [57], Landsat 7 Enhanced Thematic Mapper Plus (L7 ETM+) [58], and Landsat 5 Thematic Mapper (L5 TM) [59] sensors. These scenes were accessed using GEE [24] and represent the highest quality datasets given their Level-1 Precision and Terrain correction, typified radiometry, and well-adjusted inter sensor calibration across Landsat sensors. This col-

lection was selected due to its higher-level Tier 1 processing to atmospherically corrected surface reflectance using the USGS Land Surface Reflectance Code and Landsat Ecosystem Disturbance Adaptive Processing System algorithms. The Landsat sensors used in this study have a temporal coverage from 2000 to 2015 with a return time of 16 days at a 30 m multi-spectral spatial resolution along a 185 (115 mi) swath [24,57–59].

Using a combination of various satellite sensors increased our ability to create more cloud-free observations. Some studies have demonstrated the utility of Landsat sensor combination by increasing the probability of cloud-free image observations, while simultaneously reducing the revisit interval [60,61]. This included 2813 images from L5 TM, 8164 from L7 ETM+, and 2714 from L8 OLI. We used the deprecated Landsat collections that were available in GEE at the time of the analysis. The Landsat collections were first harmonized using a least-squares regression to maintain temporal continuity over the varying sensors in the time series [48,62]. For each Landsat image, we selected the visible (red, green, blue), near-infrared (NIR), and short-wave infrared (SWIR) bands. We calculated the NDVI and the GCVI, the latter of which has been shown to capture greater areas of vegetation given its leaf area index [9]. We then constructed annual composites between the years 2000 and 2015 by calculating the medoid, which is a more temporally robust alternative to median per-pixel compositing [62–64]. To fill any remaining data gaps, we applied a focal mean filter with a 20-pixel radius as our last step in pre-processing the annual Landsat composites.

### 2.2.3. Synthetic Aperture Radar Composites

To develop a time series of SAR data that maximized coverage of the study period, we used four global 25 m Yearly Mosaic PALSAR images from 2007 to 2010 collected by the Phased Array L-band Synthetic Aperture Radar-2 (PALSAR) aboard the Advanced Land Observing Satellite-1 (ALOS) PALSAR from the Japan Aerospace Exploration Agency (JAXA). These were accessed in the GEE data catalog [24,65], which houses the data for analysis. The collection was already ortho-rectified and slope-corrected using the 90 m SRTM Digital Elevation Model and destriped to equalize differences in intensity between adjacent strips by GEE [24,66,67]. We converted the raw 16-bit DN horizontal-horizontal (HH) and horizontal-vertical (HV) values to gamma naught values in decibel units (dB) using the equation provided by GEE [24].

To increase observations from a second SAR sensor, we used 310 images from the Sentinel-1 C-band Synthetic Aperture Radar Ground Range Detected (GRD) dataset for the year 2015, which are provided within GEE already pre-processed using the Sentinel-1 Toolbox workflow [68]. We developed annual median composites from Sentinel-1 using VV (vertical transmit, vertical receive) and VH (vertical transmit, horizontal receive) polarizations collected with the interferometric wide swath mode (IW) at a 10 m resolution. Additionally, the VH/VV ratio, as well as ratios between VV median composites, were calculated for two months (April and October) representing different seasons, calculated as VVApril/VVOctober, representing cropland seasonality [69].

**Table 1.** Overview of our variable inputs including our GFSAD basemap; Landsat ETM+, TM, and OLI; JAXA ALOS PALSAR/PALSAR-2; and Copernicus Sentinel-1.

| Variable | Spatial & Temporal Resolution | Sensor | Data Source |
|---|---|---|---|
| GFSAD Basemap | 30 m, 2015 | - | [56] |
| Blue, Green, Red, NIR, | 30 m, 2000–2015 | Landsat ETM+, TM, OLI | [57–59] |
| Normalized difference vegetation index (NDVI) | 30 m, 2000–2015 | Landsat ETM+, TM, OLI | [57–59] |
| Green chlorophyl vegetation index (GCVI) | 30 m, 2000–2015 | Landsat ETM+, TM, OLI | [57–59] |
| HH, HV | 25 m, 2007–2010 | JAXA ALOS PALSAR/ PALSAR-2 | [70,71] |
| HH/HV | 25 m, 2007–2010 | JAXA ALOS PALSAR/ PALSAR-2 | [70,71] |
| VV, VH | 10 m, 2014–2015 | Copernicus Sentinel-1 GRD | [68,72] |

### 2.3. Classification & Validation

Following the pre-processing steps, the Landsat, PALSAR-2/PALSAR, and Sentinel-1 observations were joined together in a time series stack. To enable the processing of these data over a large area, we split the study area into smaller units, using a 50 × 50 tile grid with a resolution of 25 × 22 km to divide the study extent, with 1476 of the resulting 2500 cells intersecting Zambia (Step 3 in Figure 2). Processing the data within these smaller analytical units also enabled a more replicable and flexible workflow for subsequent classification and data fusion. The 25 × 22 km resolution used here was sufficiently small to limit the computation time in GEE, enabling rapid display of intermediate results quickly, such as unsupervised classification maps and avoiding potential memory limits. Confining analyses within each tile also helped to improve performance by minimizing the amount of within-class spectral variability presented to the mapping algorithm.

### 2.3.1. Segmentation and Unsupervised Classification

The provisional classifications of Zambia were created in two stages: segmentation and unsupervised classification. We first segmented each composite into median objects using the Simple Non-Iterative Clustering (SNIC) segmentation algorithm that is available in GEE [24,41,73,74]. As an object-based processing technique, this segmentation algorithm creates pixel clusters from input information such as texture, color or pixel values, shape, and size, which is especially useful for mapping cropland [51,75,76]. The SNIC function is an improved version of the Simple Linear Iterative Clustering (SLIC) computer vision algorithm [73], making it well-equipped to run faster and more efficiently on the GEE platform. SNIC is a bottom-up segmentation approach that uses "seeds" to group neighboring pixels into clusters based on the input data and function parameters like connectivity, compactness, and neighborhood size. We used the code 'Make_SNIC_cluster' to apply the segmentation algorithm to each gridded composite, setting the connectivity parameter to 4, compactness to 1 (to enable more compact clusters), and neighborhood size to 16 pixels (to avoid tile boundary artifacts), and used a square seed pattern with a super pixel seed spacing of 8 pixels. Following the creation of the object vector boundaries, we summarized the intersecting pixels into median spectral values for each object's extent, resulting in a segmented median composite for each year and grid.

After adding spatial context into the provisional classification through the SNIC segmentation algorithm, we applied the K-means unsupervised classifier on the segmented composites to differentiate cropland and non-cropland objects. K-means is an unsupervised clustering algorithm available on the GEE platform that can be used to classify pixels or objects [77,78]. We used an input value of 20-classes in the K-means algorithm, resulting in a 20-class, segmented unsupervised classification for each of the 16 years and each of

the 1476 grids. Initial tests showed that the visible, NIR, SWIR, and vegetation indices were particularly useful for separating rainfed crop, irrigated crop, natural vegetation, and urban areas in Zambia [9], and the number of clusters (k) did not impact the cropland classification in cases where k was larger than the potential number of land cover classes. These gridded time-series data stacks were then used as input into BULC-U to distinguish cropland and non-cropland areas, as described in the next section.

To evaluate how well the unsupervised classifications distinguished cropland from non-cropland, we analyzed the overlap between the unsupervised clusters and the cropland and non-cropland classes derived from the GFSAD basemap in each tile. If the unsupervised classifier was maximally effective, the pixels in each cluster should correspond with a single land cover class (e.g., cropland), while the most ineffective classifier would contain an equal mix of all classes. Therefore, to evaluate the accuracy of the classifier, we found the proportion of the dominant class in each cluster and then calculated the mean proportion across all clusters, weighted by the size of each cluster. The results of this analysis (see Supplementary Information) reveal that the average proportion across all tiles was 92.1%, with a standard deviation of 10.1%, showing generally good separability between classes.

### 2.3.2. BULC and Crop Expansion

The BULC-U algorithm is designed to examine a time series of unsupervised classifications with the goal of extracting signals about land-use/land-cover change [40]. By comparing a given unsupervised classification's classes with those in a known higher-quality map, BULC-U uses Bayes' formula to update probabilities of LULC change through time. When quantifying the relationship between a given unsupervised class and the comparison map, BULC-U adjusts the probability of classes being tracked based on the new evidence provided by each unsupervised classification. BULC-U's developers demonstrated its ability to map LULC change in an agricultural setting with unsupervised Landsat classifications and the GLOBCover 2009 dataset, using BULC-U to refine the coarser GLOBCover map while also producing a credible multi-decadal time series of change [40].

In this study, we followed the outlines of that approach while making several modifications to address the challenges of mapping in this setting. The current state-of-the-art map for this region was the 30 m GFSAD product, which distinguishes cropland from non-cropland classes. The basemap was used to initialize BULC-U, which we ran in reverse chronological order from 2015, with the algorithm comparing the results of each unsupervised classification in each tile to the GFSAD map, and then producing a cropland probability for each 30 m pixel for each year.

To identify pixels that experienced cropland gain, we fit a linear function to each pixel's probability time series and examined the slope, using a slope-based threshold to distinguish areas of probable cropland gain from stable non-cropland areas, areas that had intermittent or unclear cropland history, or croplands that became abandoned. We tested a range of single slope thresholds to examine the relationship between the threshold and resulting accuracy. Figure 3 shows examples of each: cropland gain showing a positive slope (a, b); persistent crop and non-crop slopes that are effectively flat (c, d); cropland loss indicated with a negative slope (e), and pixels with a land use signal that alternates between crop and non-crop (f).

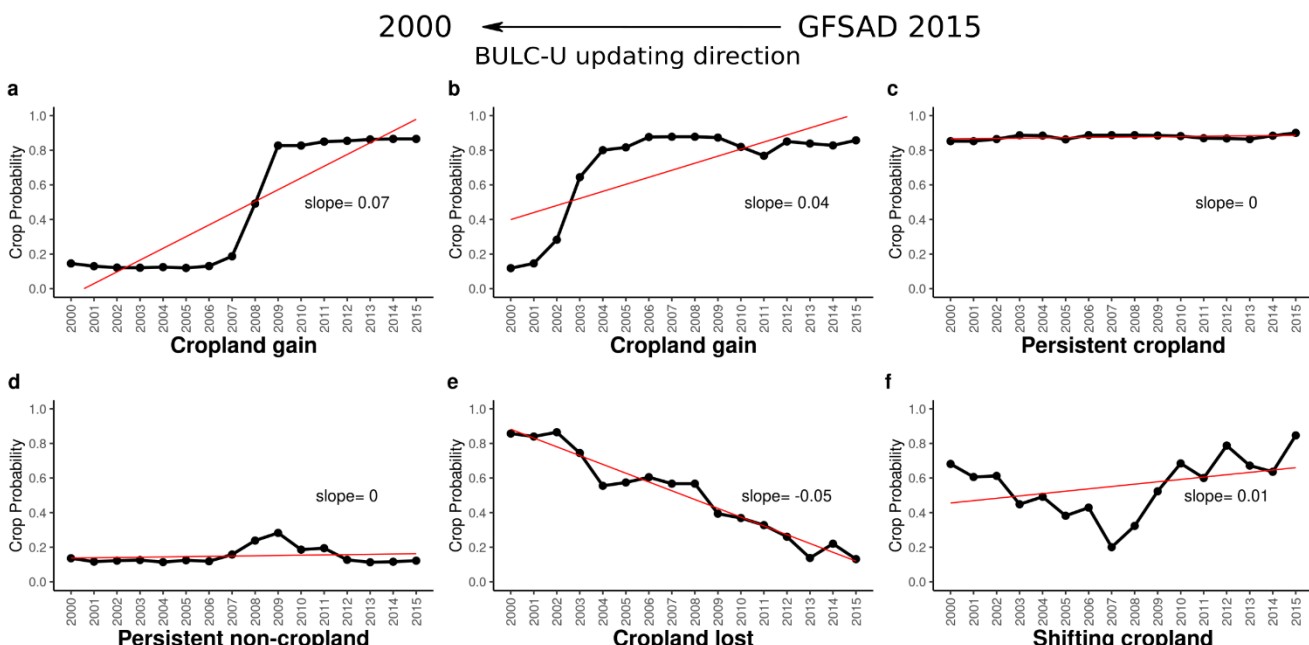

**Figure 3.** The crop class probability slope trend (black dots and line) fitted with a simple linear regression (red line) for cropland gain (**a**,**b**), persistent cropland (**c**), and persistent non-cropland (**d**), crop abandonment (**e**), and intermittent cropland (**f**).

### 2.3.3. Shapelet Analysis

Following the identification of pixels that experienced cropland gain, we identified when the gain occurred. In a previous study [69], the shapelet method was developed to detect change and classify the land cover responsible for that change based on the shape of a Landsat-NDVI time series. The "shapelet" was defined as a temporal segment within the Landsat-NDVI time series with a pattern that is highly predictive of its class. In that study [69], which detected tree plantations, the pattern was one in which there was a time period of consistently low vegetation cover (or bare ground) due to pre-planting land clearing, whereas other land uses had consistently dense vegetation cover [79–82]. That approach searched along an NDVI time series to identify shapelets characterizing intervals with low NDVI values, followed by a simple statistical test that examined whether the NDVI values in the shapelet were different than those in the time series outside of the shapelet, resulting in a land cover classification and an estimate of the year in which the plantation was established [79]. The advantage of the shapelet method is that it enables more reliable detection of change in a time series by ignoring "unimportant" time points, while improving classification results [79,80,83,84]. We adapted this shapelet method to identify the year of cropland gain for a time series of cropland probabilities, as described in the steps outlined below.

**Step 1.** For a given time series *ts* (here 2000–2015) with a yearly time step length of *l*, we arbitrarily chose a window size (*w*) of 3. The size of the window can be any value smaller than the length of the time-series but must also be greater than one and be narrow enough to suggest the year that a time series shapelet method experienced the most change.

**Step 2.** Segment the *ts* into a candidate shapelet (also referred to as *S_candidate*), and then one or two non-shapelet segments (or *N_candidate*) by moving the window from one side of the *ts* (at *t = 0*, in our case, it is the year 2015) to the end year (at time *t = l − w +1*, in our case, year-end 2000). For each time *t*, the window stands for a candidate shapelet *S_candidate* at (*t, t + − 1*), and thus segments the rest of the *ts* into one or two non-shapelet segments *N_candidate* or *N_candidate*$_{left}$ and *N_candidate*$_{right}$, depending on if the window starts from *t = 0* or is found at the end of *ts*.

**Step 3.** We selected the candidate shapelet *S_candidate* segment that was most different from its corresponding non-shapelet (s) as the final shapelet *S* using an unsupervised method for identifying and extracting shapelets [79,81], which measures the mean and standard deviation difference between *S_candiate* and the corresponding *N_candidate* to calculate an overall "gap" score that quantifies the size of the decrease of crop probability in *S_candidate*. The final selected shapelet *S* is the one that maximizes the gap score. In our study, we modified this approach by separating non-shapelets into *N_candidateleft* and *N_candidateright*, measuring their standard deviation separately because the cropland probability before and after cropland gain will have large differences, which should not be combined and measured. Then, we calculate the mean of their standard deviation, or *mean(NS_candidate_left_sd*, *NS_candidate_right_sd)* unless there is only one *N_candidate*, which indicates it is at the beginning or end of the time-series. *S_candidate_sd* and *NS_candidate_sd* values that capture the steep monotonic crop probability changes within the *S_candidate* and *NS_candidate* should be selected.

Figure 4 presents two examples of crop probability slopes and the selected shapelet windows. The first (Figure 4A) represents cropland gain that occurred later in the time series, which the maximum gap score identified as taking place in 2009, while the second (Figure 4B) shows cropland gain that was identified as occurring earlier in the time series (2003).

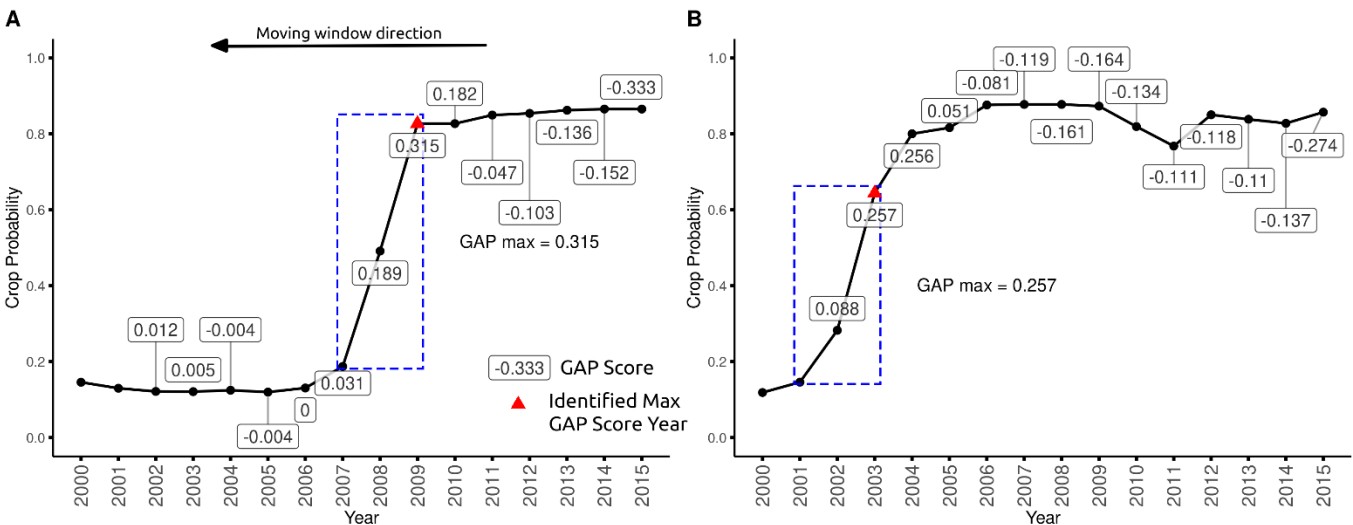

**Figure 4.** The identified shapelet (blue box) for two cropland gain examples. The GAP score was labeled for each point and the point of max GAP score was labeled with a red triangle for 2009 (**A**) and 2003 (**B**). Note the moving window starts from 2015 and moves towards 2000, following the same direction as the BULC–U update, but for clarity the *x*-axis starts in the year 2000.

### 2.3.4. Error Assessment

To assess errors in our maps, we created a reference sample following best practices for stratified random sampling and map validation [85] to select a sample of 406 points for each class (crop and non-crop) over the study area. Using the Collect Earth land monitoring platform [86] three experienced workers inspected each point using historical Google Earth imagery [87] that was collected on or near the years 2000, 2010, and 2015, providing a label for each class and time period. Points were labeled as cropland when more than 50% of a 90 m × 90 m square centered on the sample point was covered by agricultural fields as cropland, or otherwise labeled as non-cropland.

The labeling task was difficult because the earlier years were primarily covered by lower resolution Google Earth imagery, which made it challenging to distinguish cropland from non-cropland, particularly when there was low contrast between land covers. The labels (crop or non-crop) assigned by different workers to the same point therefore frequently disagreed, and there were several possible methods for reconciling these disagreements when assigning a final class for each point and time period, ranging from full agreement to majority agreement, or by assigning the label from the lowest frequency class if it was

selected by at least one worker. In this case, the lowest frequency class is cropland, which represents approximately 10–15% of total area in Zambia [5,88]. Each approach has its advantages and disadvantages and will lead to different estimates—and thus different understanding—of map accuracy; complete agreement increases confidence but shrinks the total pool of reference samples (if one worker disagrees, the sample is excluded), while majority agreement preserves the sample size, but may bias the sample towards the most dominant and easiest to recognize class (non-cropland in this case). The last approach, assigning the cropland class if at least one labeler selected it, which we refer to here as "cropland wins", improves representation of the cropland class at the potential expense of increased commission error (i.e., non-cropland classified as cropland). For this study, we use the last two approaches, majority agreement and cropland wins, and report results from both. Further details on how the label methods vary in their impacts on map accuracy assessment can be seen in Figure S1.

We used the resulting reference sample to conduct three different error assessments, the first of which was to evaluate the overall User's (73%), and Producer's (89%) accuracy of the 30 m GFSAD base map in 2015 as well as the BULC-U created cropland map for the year 2000. The second and third assessments respectively evaluated the ability of our method to accurately detect cropland change and to estimate the year of cropland gain. To conduct these last two assessments, we estimated the total cropland gain, loss, and persistence between three different time intervals (2000–2010, 2010–2015, 2000–2015), using the reference classes to assess the class transitions between each time interval based on the quantity of samples in each class (Table 2). The sample revealed a net cropland expansion of 11.3% between 2000–2010, and 2.9% between 2010–2015, and 14.7% overall between 2000–2015. To estimate how accurately the map identified cropland gain, we calculated the percentage of map-predicted cropland change pixels that corresponded to reference sample-identified cropland change, as well as the percentage of mapped cropland change pixels that were identified as non-change points in the reference sample. Samples that experienced cropland change were those that were non-cropland in 2000 but became cropland by 2015, while those corresponding to no change were samples that were persistent cropland or non-cropland in 2000, 2010, and 2015. To estimate how accurately the modeling approach estimated the date of cropland change, we grouped the reference samples into classes of persistent cropland, persistent non-cropland, early cropland gain (2000–2010), and late cropland gain (2010–2015) and used these points to extract their corresponding estimated year of change from the shapelet analysis.

**Table 2.** Validation points and label distributions grouped according to year of observation, and by intervals defining different change periods.

| | Type | 2000 | 2010 | 2015 | 2000, 2010 | 2010, 2015 | 2000, 2015 | 2000, 2010, 2015 |
|---|---|---|---|---|---|---|---|---|
| **Cropland wins** | Cropland | 324 | 358 | 368 | - | - | - | - |
| | Non–cropland | 488 | 454 | 444 | - | - | - | - |
| | Persistent cropland | - | - | - | 302 | 343 | 299 | 296 |
| | Persistent non-cropland | - | | | - | 432 | 429 | 419 | 410 |
| | Cropland gain | - | - | - | 56 | 25 | 69 | - |
| | Cropland lost | - | - | - | 22 | 15 | 25 | - |
| **Majority agreement** | Cropland | 205 | 256 | 271 | - | - | - | - |
| | Non–cropland | 607 | 556 | 541 | - | - | - | - |
| | Persistent cropland | - | - | - | 187 | 238 | 183 | 178 |
| | Persistent non-cropland | - | | | - | 538 | 523 | 519 | - |
| | Cropland gain | - | - | - | 69 | 33 | 88 | - |
| | Cropland lost | - | - | - | 18 | 18 | 22 | - |

## 3. Results

We ran the entire workflow (Figure 1) for each of the 1476 tiles covering Zambia with the goal of identifying cropland expansion and the year in which it occurred. We begin by illustrating the analytical steps of this process within an example grid in western Zambia that experienced cropland expansion between 2000–2015 (Figures 5 and 6). This example illustrates results for 4 of the 16 years (2000, 2005, 2010, 2015) for this tile. The high-resolution imagery and Landsat composites (Figure 5, rows 1 and 2) show the locations of cropland during the four-time intervals, as well as the variability of the vegetation condition captured by the imagery between years. The third row details the segmentation/unsupervised classification results, which shows the variability of the cropland and non-cropland areas, while row 4 shows the probabilities calculated by the BULC-U algorithm. The slope of the linear regressions fit to the time-series of BULC-U probabilities in each pixel is illustrated in Figure 6 row 1 (column 1), along with identified cropland gains from three different slope thresholds (columns 2–4), followed by shapelet results (Figure 6 row 2) illustrating the year of cropland change.

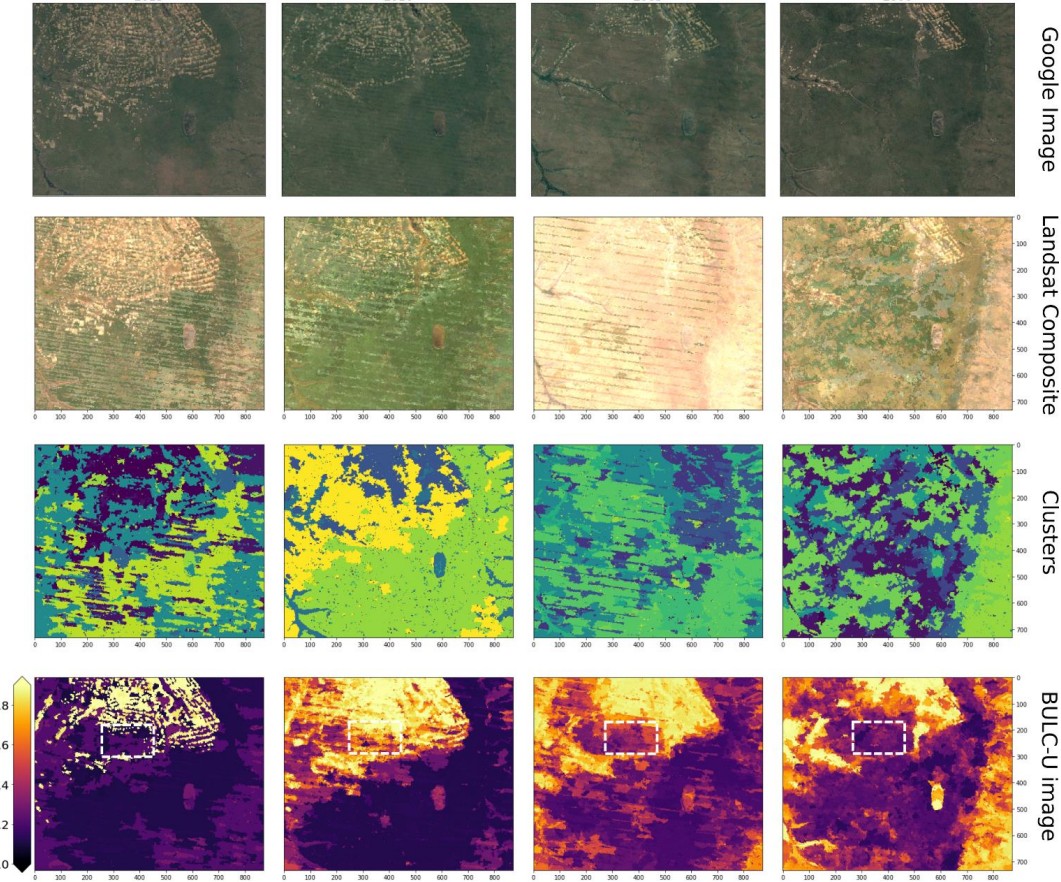

**Figure 5.** An example of the workflow output for four time points (2015, 2010, 2005, and 2000). For each time point, the Google image (top row, for reference), Landsat composites (second row), unsupervised classification with segmentation (third row), resulting cropland probabilities from the BULC-U algorithm (fourth row) are shown. The white locator box highlights high probability cropland that appears in 2010 and disappears in 2015, indicating a probable loss of cropland area.

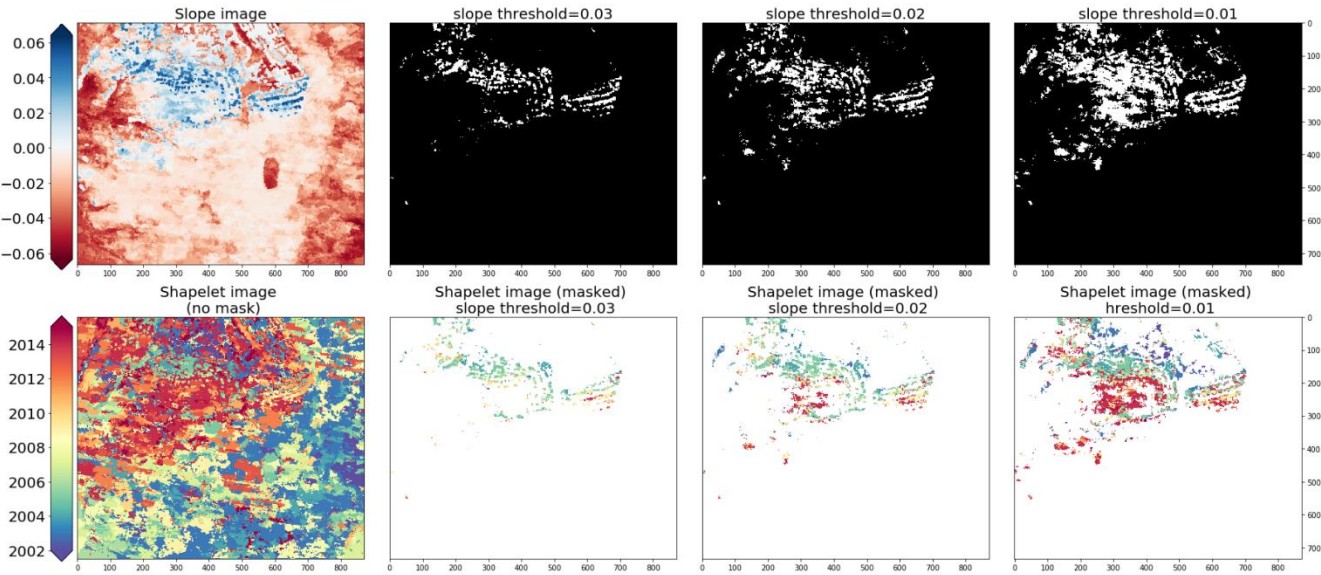

**Figure 6.** Post BULC-U analysis to identify cropland expansion and the expansion year between 2000 and 2015. The first row illustrates the calculated slope of probability values (column 1) along with the areas of cropland gain identified by using different slope thresholds (columns 2–4). The last row is the shapelet estimated cropland gain year without a mask (column 1) and with a mask applied (columns 2–4) using masks from the first row.

For context, it is important to remember that the BULC-U procedure was started in the year 2015 with the cropland and non-cropland classes in the GFSAD basemap set to probabilities of 0.8 and 0.2, respectively. The 2015 BULC-U image therefore shows a distinct contrast between low and high crop probabilities at the beginning of BULC-U updating (Figure 5 row 3 column 1). The central northern area shows a clear pattern of high crop probabilities with certain areas distributed into rows (likely associated with subsistence row crop farming). The majority of the 2015 BULC-U probabilities are low, with some areas close to zero, indicating a surrounding no crop area. The central portion (dark purple) of the tile had the lowest crop probabilities, while the most eastern, western, and southern edges of the tile show slightly higher cropland probabilities (lighter tone of purple). The land cover in the bottom outer edges of the tile is likely associated with grasslands or open canopy forest, which are spectrally similar to cropland. It may also be attributed to the unsupervised classification producing clusters that were associated with both non-cropland and cropland areas. In the year 2010, the extent of higher cropland probability pixels increased, while the non-crop areas changed little in comparison to 2015. Then in 2005 and 2000, a similar distribution of crop probability pixels is still evident, with higher crop probabilities occurring in the areas that were previously consistently low. The white locator box overlaid on the BULC-U image highlights where the GFSAD basemap had an omission error in the cropland class. This patch had low cropland probabilities (purple) in 2015, then transitioned to high probability (yellow orange) in 2010, before reverting to low probability in 2000. This transition indicates cropland that was initially missed by GFSAD but was captured in 2010 given its spectral signature of cropland in the years from 2015 to 2010, although the 2010 Google Image suggests a non-cropland area. Then, this cropland loss was captured in early years, indicated by the decrease of the cropland probability in the 2005 and 2000 images. In contrast, the east and southwest edges of the image, which were not cropland at any point in the study, showed increasing cropland probability from 2010 back to 2000, which reflect the spectral similarities between cropland and shrublands (a term we use here to encompass savannas, woodlands, and other non-cropland vegetation types with varying degrees of woody vegetation cover), which was particularly pronounced in 2005 and 2000. Using the GFSAD cropland class as a mask helped to remove such areas of

falsely detected cropland, thereby helping to reduce spurious detections of cropland and cropland change (Figure 5, row 3). However, this masking excluded cropland areas that were missed by GFSAD.

Following the BULC-U probability outputs, the slope of cropland probabilities was calculated for each image pixel across the 15 annual probability values (Figure 6, row 1, column 1). The region falling within the GSFAD 2015 mask had slope values ranging from near zero (white) in areas where the cropland was present since 2000, such as the north-central region, to positive values (blue) where cropland expansion occurred closer to 2015 (e.g., towards the northwest). Near-zero slope values are also seen in the central region because of the persistence of non-cropland between 2000–2015. Negative slopes are evident in the east and west due to cropland probabilities decreasing from 2000 to 2015, which reflects the false detections of cropland in earlier years. Adjusting the slope threshold alters the amount of cropland expansion identified (Figure 6, row 1, columns 2–4). As the slope threshold moves closer to zero, more pixels are identified as cropland gain.

The steps illustrated in Figure 5 were performed within each tile in the grid, followed by the application of the shapelet method to detect the year of change. Figure 7 illustrates the resulting cropland gain map for all of Zambia, along with the shapelet-estimated year. The results show that cropland gain was widely distributed throughout the country and steadily expanded within the vicinity of existing croplands over the 15-year time period. Visual observation indicates our methods appeared to be effective in delineating individual fields or clusters of fields, although some edge effects do exist (e.g., in the northeastern part of the lower inset in Figure 7).

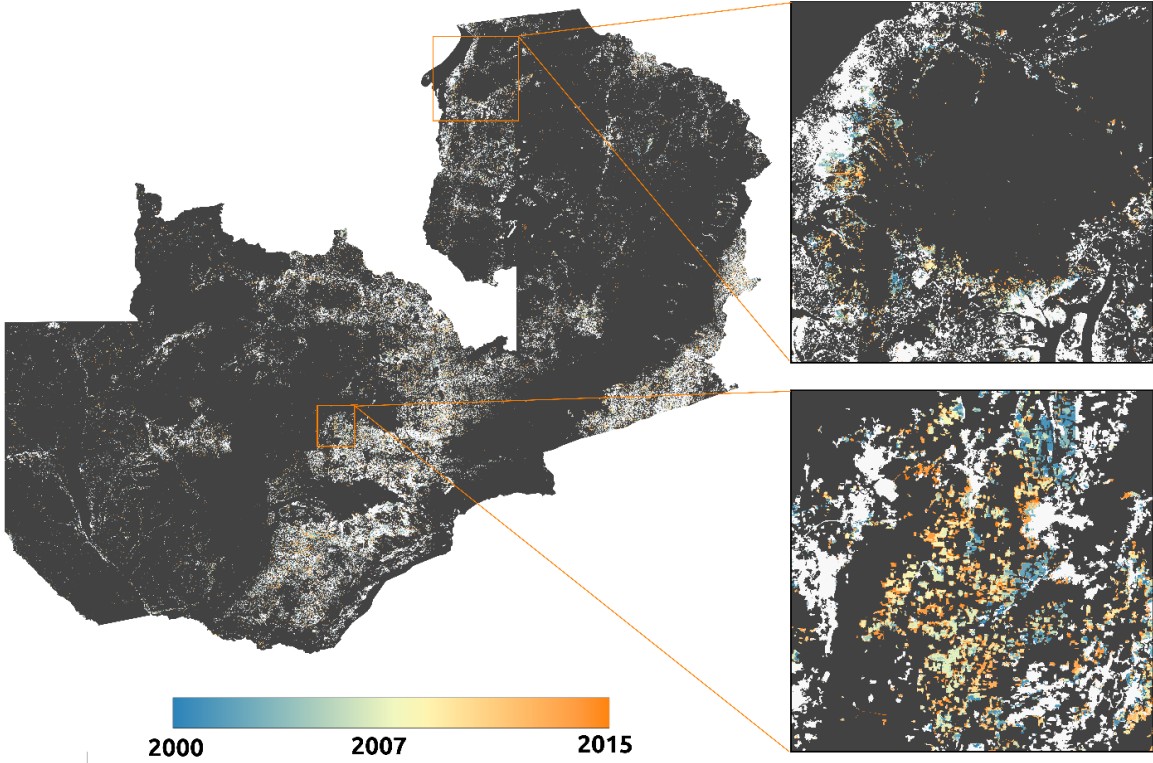

**Figure 7.** The areas where cropland expanded in Zambia between 2000–2015, color-coded by the year of expansion, shown in relation to cropland established prior to 2000 (white), as mapped by the GSFAD cropland layer.

The trajectories of the probabilities at reference sample points (Figure 8) provides further quantitative insight into BULC-U's ability to distinguish cropland gain events (e.g., expansion between 2000–2010 and between 2010–2015) from areas representing persistent cropland or non-cropland. As the BULC-U process began in 2015 and ran backwards towards the year 2000, cropland probabilities in the persistent non-cropland class were

expected to start at low values that remained low for all 16 years and persistent cropland to have high values that remained high. In contrast, the two cropland gain classes that were expected to have high probabilities that became low either between 2000–2010 (early cropland gain) or between 2010–2015 (late cropland gain). In our assessment, which was conducted for our set of validation points, these patterns generally held, with exceptions indicating model error and the relatively small size of the samples in the two gain classes, as well as variability related to the method used to develop the reference sample.

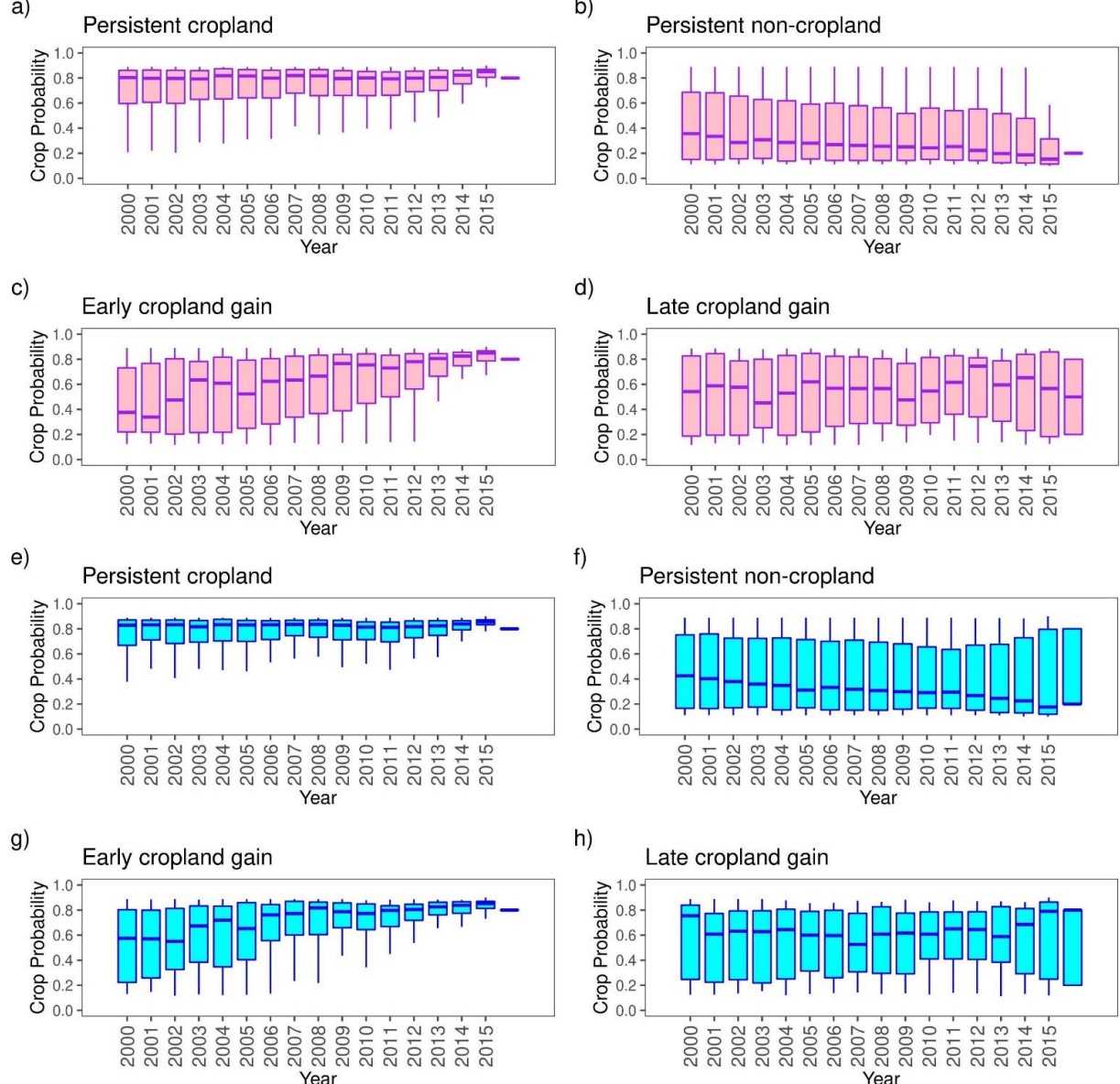

**Figure 8.** Probability trajectories within four change-based classes in cropland wins reference dataset: (**a**) Persistent Cropland; (**b**) Persistent Non-Cropland; (**c**) Early Cropland Gain (expansion between 2000–2010); (**d**) Late Cropland Gain (expansion between 2010–2015). The probability trajectories within the same four classes, as assessed with the majority agreement reference dataset, are shown in subplots (**e**–**h**). Cropland probabilities are summarized as box plots for each year in the time series, with the bar indicating the median probability in each time period, the upper and lower ends of the box showing the 75th and 25th percentile of probabilities, and the whiskers showing 1.5 times the interquartile range. The initial probability distributions on the right side of each plot are drawn from the probability value assigned to the two GSFAD classes (0.8 for cropland, 0.2 for non-cropland) prior to BULC-U updating.

For example, under the cropland wins reference sample, most persistent cropland (Figure 8a) started from high cropland probabilities in 2015, and as updates proceeded towards 2000, many cropland pixels retained high probability with the median annual probability stable at ~0.8. However, the 25th percentile probability value gradually decreased from 0.8 to 0.6, with the range of values below the 25th percentile also gradually declining to below 0.2, which indicates a general decrease in cropland probabilities within the stable cropland class from 2015 towards 2000. The persistent cropland class showed the same pattern when using the majority agreement reference set (Figure 8e), albeit with a narrower range and smaller reduction in the probability values.

The persistent non-cropland (Figure 8b,f) showed a similar tendency under both reference datasets, beginning with low cropland probabilities in 2015 (median = 0.2) that remained low at all time points, but with slight increases towards a median of 0.4 as updates moved towards 2000. In contrast to the persistent cropland class, however, the distribution of cropland probabilities in the non-cropland class was much wider in each year (Figure 8f), with the 75th percentile extending between 0.3 and 0.7 (up to 0.8 under the majority agreement reference set), suggesting that there were numerous commission errors within the persistent non-cropland class. These errors were most pronounced in 2015, reflecting the GFSAD's tendency to over-predict cropland. The high initial cropland probabilities in the persistent non-cropland class (Figure 8f) might also be due to the potential omission error in our majority agreement reference dataset since the requirement for majority agreement rejects more cropland. It is therefore possible that GFSAD correctly classified such areas as cropland and were placed into the persistent non-cropland class because of misclassification in our reference labels. The increases in the median values towards 2000 might be explained by the mix of cropland and non-cropland in the unsupervised classification.

The early cropland gain class (Figure 8c,g) had an initial high cropland probability in 2015 and an overall decrease of median cropland in both cropland wins and majority agreement reference dataset. For the cropland wins reference dataset (Figure 8c), the most substantial decreases in median cropland probabilities were between 2009 to 2005, and 2003 to 2000. For the majority agreement reference set (Figure 8g), the most substantial decreases occurred between 2008 to 2005 and 2003 to 2000. The declining median values were accompanied by a substantial increase in the interquartile range (IQR, the range between the 25th and 75th percentiles), which may indicate the difference in years in which cropland to non-cropland transitions were detected as BULC-U updated towards 2000. However, nearly 25% of the data in this class (upper whisker) had cropland probabilities that were persistently greater than 0.8, although these cropland probabilities also decreased slightly, given the decline in the 75th percentile towards 2000.

The late cropland gain class (Figure 8d,h) had an initial median probability of 0.5 when using the cropland wins validation sample (Figure 8d), with a relatively wide IQR (0.2 to 0.83). For the majority agreement validation sample, the median initial probability was 0.8 (Figure 8h), with the higher value reflecting the fact that the majority agreement sample had a greater correspondence with GFSAD's cropland class. Under the cropland wins reference set, the increase in the median cropland probability between the initial condition to ~0.6 in 2014, going as high as ~0.8 in 2012, could indicate the repair of initial false negative error (cropland missed by GSFAD) by BULC-U, while the declines in probabilities from 2012 to ~0.5 in 2009 shows the system detecting post-2010 cropland gains. However, there is no clear trend in earlier years, and the number of observations in this class is relatively small, which contributes to the noise. For the majority agreement reference dataset, the cropland probability dropped from the initial condition of 0.8 to 0.6 in 2013, then increased a little to ~0.65 and stabilized until a sudden rise in 2000.

The percentage of cropland gain and the corresponding false positive error that occurred between 2000 and 2015 varied according to the slope threshold that was used, (Table 3 and Figure 9). For example, using a threshold of 0.03 captured 27.5% of the cropland gain in estimate using the reference sample, with a false positive rate of 6.9%. Since the update is based on the basemap (GFSAD), if we exclude the validations points falling

outside the basemap, then this threshold captures 35.8% of reference cropland gain, with a false positive rate of 13%. As the slope threshold moves towards zero, the amount of cropland gain captured increases, but at the cost of a higher false positive rate. However, between the thresholds of 0.03 and 0.01, the gains in percentage of cropland gain captured increased from just over 25% to nearly 51%, while the overall false positive rate remained less than 18%. Lowering the slope threshold from 0.01 to 0.005 increased the amount of cropland gain captured to 70% and increased the false positive rate to 38%. The high commission error when evaluating cropland gain confined to the base map was caused by the relatively high commission error in the base map together with the improvements in map accuracy resulting from BULC-U updates. In other words, areas falsely classified by the basemap as cropland in 2015 had their probabilities reduced by BULC-U in earlier years, which caused these areas to be falsely identified as cropland gain.

**Table 3.** The percentage of actual cropland gain (per the reference sample, refer to Section 2.3.4) captured by the mapped cropland gain by slope threshold, along with the commission error (commission error represents areas that did not have cropland gain but were mapped as such).

| | Slope Thresh | Cropland Gain Captured (Zambia) | Commission Error (Zambia) | Cropland Gain Captured (GFSAD 2015) | Commission Error (GFSAD 2015) |
|---|---|---|---|---|---|
| **Cropland wins** | 0.005 | 0.696 | 0.37 | 0.736 | 0.598 |
| | 0.01 | 0.507 | 0.178 | 0.623 | 0.329 |
| | 0.015 | 0.435 | 0.144 | 0.547 | 0.269 |
| | 0.02 | 0.406 | 0.108 | 0.509 | 0.198 |
| | 0.025 | 0.348 | 0.084 | 0.453 | 0.158 |
| | 0.03 | 0.275 | 0.069 | 0.358 | 0.13 |
| **Majority agreement** | 0.005 | 0.705 | 0.36 | 0.77 | 0.596 |
| | 0.01 | 0.5 | 0.166 | 0.581 | 0.316 |
| | 0.015 | 0.466 | 0.132 | 0.554 | 0.254 |
| | 0.02 | 0.42 | 0.099 | 0.5 | 0.187 |
| | 0.025 | 0.375 | 0.073 | 0.446 | 0.143 |
| | 0.03 | 0.33 | 0.06 | 0.392 | 0.117 |

Finally, to evaluate the accuracy of the estimated year of cropland gain, we intersected the reference labels for early (2000 to 2010) and late (2010 to 2015) gain with the years estimated from our shapelet method, excluding cropland gain labels that were omitted by our slope method. Figure 10 (lower) shows the histogram of shapelet-detected gain years for both cropland gain classes (early and late), corresponding to reference labels developed using the majority agreement strategy (see Section 2.3.4). Although there is substantial overlap in the distributions of gain years between each class, and only 11 observations in the late gain class, 55% of the detected years in the late gain class were after 2010 while 77% of the values for the early class were before 2010, with the mode at the year 2003. Both percentages are larger than the outcomes expected purely by chance, which would be 33% of detected gain years in each 5-year bin (with 67% in the two pre-2010 bins). Results of the same analysis based on the reference labels developed using the cropland wins strategy (see Figure 10 upper) showed lower performance for the late gain class, with just 44% of the late gain class correctly identified, although this sample size (8) was even smaller. The early gain class was more accurate, with 69% of the sample correctly detected, although the model value (7, or 22%) occurred around 2014.

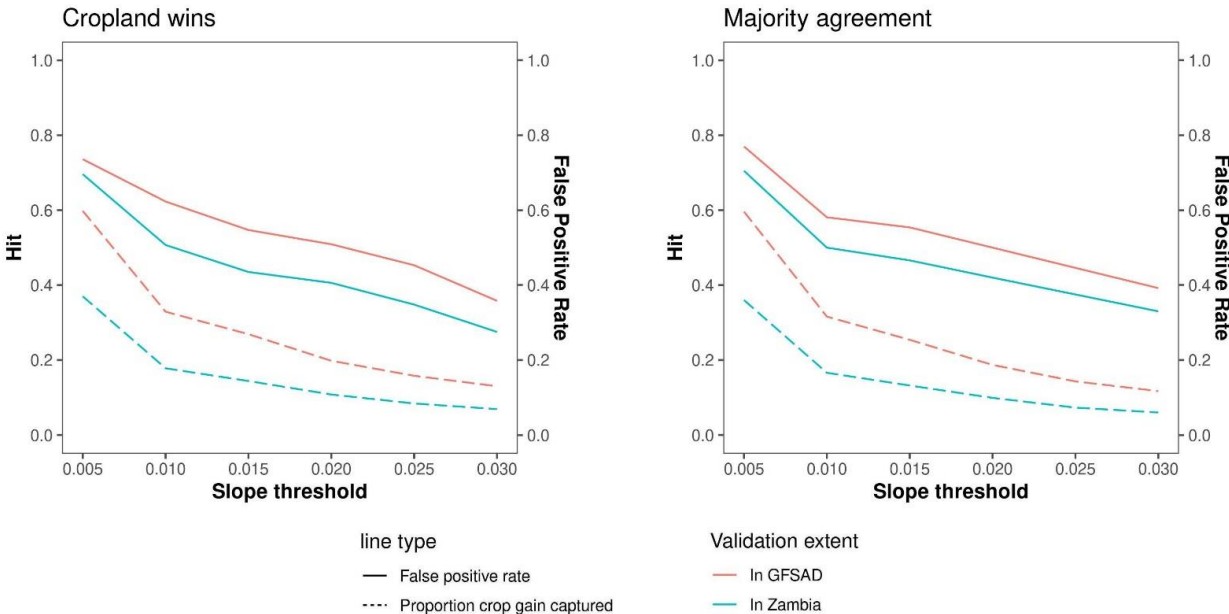

**Figure 9.** The percentage of actual cropland gain captured by the mapped cropland gain, as determined by varying slope thresholds, along with the commission error (reference samples points that showed no cropland gain but were mapped as cropland gain by BULC-U for the reference sample transition class).

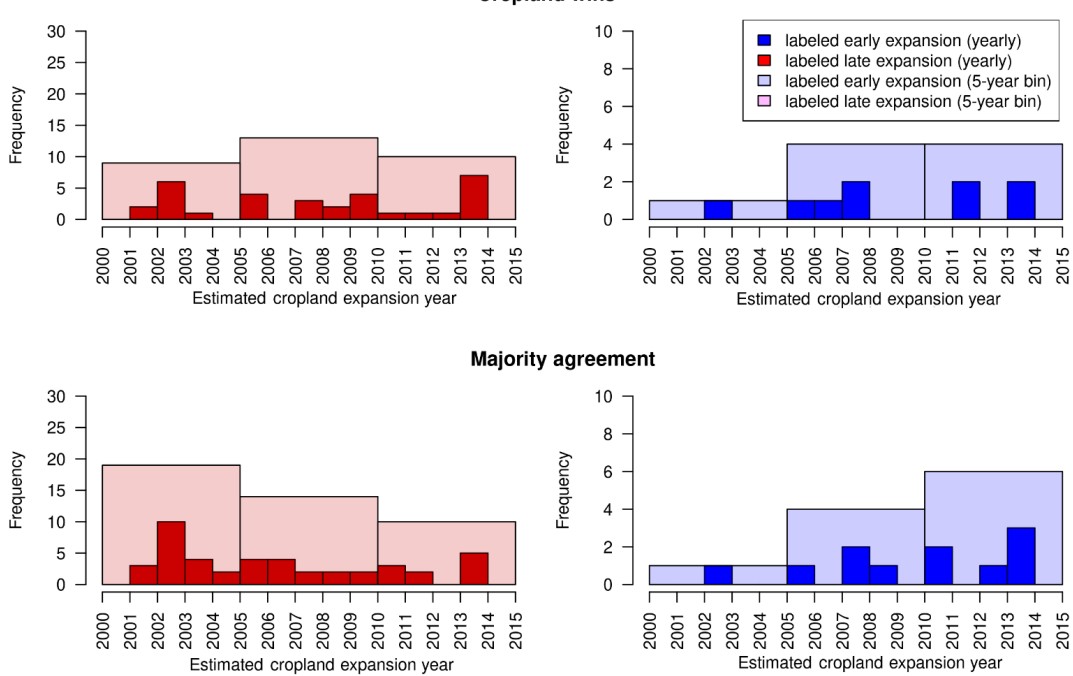

**Figure 10.** Reference labels (based on the cropland wins (upper) and majority agreement (lower) strategy to determine label class) of early cropland expansion (2000 = non-crop, 2010 and 2015 = crop) and late cropland expansion (2000 and 2010 = non-crop, 2015 = crop), and their corresponding estimated cropland expansion year (*x*-axis) from the shapelet method. A slope threshold of 0.005 was used to detect cropland expansion, to maximize the amount of cropland gain detected. The histograms do not include cropland expansion events that the method failed to detect. Darker bars indicate the number of gain events in individual years, while lighter bars summarize the number of events in five-year intervals.

## 4. Discussion

This study demonstrated a new capability to track agricultural change at medium resolution over large areas by mapping annual cropland expansion over 16 years in Zambia, a country that covers 750,000 km$^2$. To enable this capability, we leveraged and extended the large-scale processing capabilities of GEE and the Bayesian-synthesis approach provided by the BULC-U algorithm, which we trained using a single high quality cropland base map, thereby removing the need to collect extensive training data. We applied this method to annually composited Landsat Surface Reflectance and synthetic aperture radar (Sentinel-1 and ALOS PALSAR) data at 30 m resolution, which improved the depth and variety of information available to the classifier, while helping to overcome the limits to observation posed by frequent cloud cover. Applying regression and shapelets to the BULC-U-generated cropland probability time series enabled us to identify where and when cropland expansion occurred. Using validation protocols for change detection (cropland gain versus non-cropland gain) and land-cover classification (cropland versus non-cropland) accuracy, we found that this approach showed fair to good performance in detecting cropland expansion events in a hard-to-map environment, while helping to improve the accuracy of annual cropland maps. Although we confined our analysis to cropland expansion, which is by far the dominant mode of agricultural land cover change in this region [5], our method can be adapted to analyze other cropland change patterns (e.g., loss, shifting agriculture patterns). This method complements similar recent global-scale efforts [89] by providing an ability to map cropland change at higher temporal frequencies, which can provide additional opportunities for understanding the drivers and impacts of those land cover changes.

In the following sections, we provide a detailed discussion of the effectiveness of the individual components of our approach, including suggestions for further improvements.

### 4.1. Factors Impacting Classification Accuracy

The complexity and diversity of Zambian croplands, together with the spatial-spectral similarities between croplands and shrublands, caused confusion in our provisional classifications, especially for the annual Landsat mosaics. Distinguishing between these two classes with 30 m Landsat TM imagery can be challenging [10], and we saw this error reflected in the overall increase in cropland probability in areas covered by shrublands, as highlighted in Figure 5. This confusion was caused by the inability of the unsupervised classification to differentiate cropland from shrubland. Beyond those errors, there was also confusion between natural vegetation and small-scale crop fields, which often have remnant large trees and indistinct boundaries, or may be fallow [90]. There are opportunities to reduce this misclassification by including predictor variables that can further help distinguish cropland from non-cropland, such as seasonal NDVI composites [91] and a spatial index [92].

In this study, we used annual median composites because the lack of cloud-free observations prevented us from making seasonal composites, especially during the growing season, as is common in many agricultural regions [43,93]. As seasonal composites provide better separability between croplands and non-croplands [51,92], future studies could focus on integrating observations from other sensors, such as Sentinel-2 or PlanetScope, in order to create higher temporal resolution composites that can provide this seasonal contrast. The moderate spatial resolution of Landsat is too coarse in some instances to classify smallholder cropland, as the average size of crop fields in such systems is <0.64 ha [94], which underscores the value of integrating higher spatial resolution data into the composites. Further gains may be obtained by incorporating data from active sensors, as we found that SAR backscatter helped to increase observations in cloud-heavy regions while providing complementary information that helped to distinguish croplands from other land covers [95]. Unfortunately, there are limitations to accessing raw SAR observations prior to the launch of Sentinel-1 in 2014, because data from other SAR satellites are not publicly accessible or have limited availability on the GEE data catalog. For example, the ALOS PALSAR data we used in this study that was shared on GEE are not the raw

observations but rather a yearly mosaic, and ALOS PALSAR is only available for the years 2007–2010 [24,66].

*4.2. BULC-U for Tracking Cropland Change*

The unsupervised, object-based classifications from annual multi-sensor observations were effectively fused together by BULC-U to construct annual cropland maps for Zambia. In this study, we focused on a binary classification (cropland and non-cropland) problem, but the BULC/BULC-U algorithm can be extended to multiple land-cover classes. Future work could track multiple land cover conversion pathways using the workflow that we have developed, with minor input parameter changes in BULC-U.

BULC-U was able to integrate additional evidence from the input classifications to improve upon the quality of the base map when calculating the initial class probabilities. The GFSAD we used to initialize BULC-U missed some smaller regions of croplands (Figure 5), but as the update events were synthesized in BULC-U, the omitted cropland was detected by the unsupervised classifications and the cropland probabilities increased in the direction of BULC-U updates (backwards in time). This ability to improve the accuracy of an existing map is one of the features of BULC-U [40] and could be further improved by increasing the frequency of updated events (e.g., by applying unsupervised classifications to seasonal image composites), or by running BULC-U backwards and then forwards again [44]. Another means for improvement is to start with a more accurate base map, which should in turn enhance BULC-U's ability to map cropland gain. The accuracy of the base map is crucial because it was used to calculate the initial cropland probability against the first provisional classification input. Based on the comparison of the GFSAD with our reference data, the GFSAD had a high producer's accuracy (89%) but a lower user's accuracy (73%). In terms of our results, these base map errors were propagated into our dataset through the initialization of the cropland and non-cropland probabilities (Figure 8). As a result, most cropland was correctly assigned with a high initial cropland probability (Figure 8, lower-right) in BULC-U because its high producer's accuracy of GFSAD meant that it was less likely to miss cropland. However, some non-cropland pixels were initialized with high cropland probability because of the relatively low user's accuracy of cropland in GFSAD (non-cropland was mapped as cropland). Therefore, future studies could explore using a basemap with higher accuracies to create more robust initial probabilities, and even potentially interspersing multiple basemaps within the time series. As mentioned in the previous section, newer sources of imagery together with improved classification algorithms, particularly neural networks [34], offer the possibility of generating improved basemaps [96,97].

*4.3. Benefits of Gridded Processing Units*

In addition to being necessary for processing a long time series of imagery over a large area, we found that there were additional advantages to using a smaller gridded processing unit in mapping workflow. First, the computational load scales linearly with the study area, thus the smaller unit enables rapid processing of the imagery, followed by application of BULC-U on the data stacks, and then downloading the resulting probabilities for visualizing or post-processing on a local computer. This ability to rapidly analyze and then re-run improves accuracy by allowing more permutations to be assessed and helps to catch coding and analytical errors. Second, the relatively small study area provided by the tiled grids were less impacted by the quality of radiometric correction or the atmospheric conditions across the processed image's extent [98]. Although advances in Landsat data processing have improved the consistency of the radiometric response over the land surface within and across Landsat scenes [99], as with the LEDAPS [100] surface reflectance product, radiometric variability over large areas remains a problem that can influence results. Using smaller processing units helped to limit the impacts of such radiometric bias in the resulting classifications of annual Landsat mosaics and subsequent Bayesian data fusion. Third, a smaller processing unit helped to reduce regional errors from low-quality unsupervised

classifications or inaccurate reference map data. By using the gridded system, such errors were isolated within their own grids, limiting their influence on the BULC-U algorithm. Furthermore, the gridded processing system enables the application of local modifications to the algorithm, such as varying thresholds, which can help to further improve overall accuracy. However, future applications could explore ways to limit edge effects (e.g., by building in and then averaging across tile overlaps), as we found from our uniformed slope thresholding. While we ran BULC-U using 25 km by 22 km gridded tiles in our study, we found that using larger grid sizes is also feasible.

This gridded processing system can also support the use of other advanced data fusion and classification methods available in GEE, which may also have high memory and computing resource requirements and would not complete if the study area is too large.

### 4.4. The Effectiveness of the Threshold and Shapelet Method

We found that the slope-based threshold method was effective for identifying cropland gain given the clear boundaries of cropland in this region (Figures 5 and 7). Instead of choosing a final slope threshold in this study, we tested a sequence of slopes experimentally to allow us to identify the best threshold for the given features. Future studies could use a subset of our cropland and non-cropland regions to identify the ideal slope that maximizes the separation of cropland and non-cropland samples.

The shapelet method, which was previously shown to be effective for detecting the year of tree crop plantings [79], proved useful for estimating the year of cropland expansion. In our study, we applied multiple methods to evaluate the effectiveness of the shapelet method using historical imagery from Google Earth, visual interpretation of example sites, and reference points for the validation of the entire results. From the example sites, we found that the shapelet method was generally able to capture the relative chronological order of cropland expansion. Figure 10 also gave an estimation of the effectiveness of the shapelet method, indicating it can distinguish between earlier and later cropland expansion, although the number of gain observations in our reference sample was too small to precisely measure how well the timing of gain events were captured. Future work should increase the sample size within gain classes and could apply annual validation to further evaluate the shapelet method's strength for identifying cropland gain. In addition to improving classification accuracy, adding more frequent intra-annual observations to the time series could also improve the shapelet method's effectiveness. Finer time intervals would allow the shapelet method to identify cropland gain in a window of one year instead of three years. Detecting multiple shapelets could also help to identify fields in areas of shifting cultivation (e.g., in Northern Zambia) within the cropland probability time series. To achieve this, both the direction and magnitude of cropland probability change would need to be used to detect multiple shapelets.

### 4.5. The Impact of Reference Label Uncertainty on Understanding Performance

An important point to note regarding our reported results is that the range in assessed classification accuracy and change detection performance in large part reflects the varying levels of confidence within the reference label classes used (see Section 2.3.4). The uncertainty in the reference labels arises from the fact that they were developed through visual interpretation of imagery that was of varying, and primarily moderate, resolution. Identifying smallholder cropland within moderate resolution imagery can be difficult, resulting in higher labeling error and greater between-labeler disagreement. The reliability of our reference sample was therefore lower in the earlier years of our study period (see Figure S1), when the image archives are primarily composed of Landsat data. That uncertainty, and the way it was handled when aggregating the sample, had substantial impacts on the performance metrics for each of the three assessments.

Another limitation of our reference data is that it only covered three time points, which limited our ability to assess how well our methods can detect annual change events. These factors demonstrate that the lack of high-quality reference data remains one of the major

hurdles to land cover change mapping [101,102]. A fuller assessment of the performance of our methods will therefore require a higher quality reference dataset, as the accuracy of the reference labels sets the upper bound on knowable map accuracy.

### 4.6. Detecting Other Patterns of Cropland Change

Our current method is limited to detecting cropland gain events, but the probability time series generated by BULC-U provides the ability to detect other patterns of cropland change, such as cropland abandonment, or cycles of gain and loss that commonly occur in swidden agricultural systems. Detecting other, more complex events (e.g., cropland gain followed by cropland loss) will require methods that can recognize change points within more complex, polynomial functions. Therefore, more sophisticated temporal segmentation algorithms could be used, such as the trajectory detection method described in LandTrendr [64], or approaches based on continuous change detection and classification [103,104].

### 5. Conclusions

In this study, we demonstrated how to use a flexible GEE-based workflow using BULC-U, slope-thresholding, and shapelets to identify cropland expansion between 2000 to 2015 across a large region experiencing rapid agricultural growth, in this case Zambia. By initializing the workflow with a single existing land cover map, our approach also demonstrates a potential solution for overcoming the challenge of limited training data, while showing how optical and SAR data can be combined to improve the frequency and depth of information needed to effectively distinguish cropland from other land cover types in a cloudy region. Future opportunities exist to build upon the methods presented here by using more accurate land cover data to initialize the workflow, and by integrating a more sophisticated time-series shape detection method. Ultimately, our flexible approach makes mapping cropland change dynamics across large areas more accessible, further advancing opportunities for cropland monitoring on cloud-based processing platforms.

**Supplementary Materials:** The following supporting information can be downloaded at: https://www.mdpi.com/article/10.3390/rs14194896/s1. Figure S1: Label consistency among three workers. First row: Crop labels. Second row: non-crop labels. Note: circle sizes were scaled in each subplot and were not comparable to other subplots. Figure S2: The consistency among three workers in the created crop labels. Worker A has a smaller crop label size in all three years compared with workers B and C (first row). Three candidate strategies were considered to unify the validation labels as vote (second row), favoring crop (third row) and favoring non-crop (last row). Figure S3: Cluster Purity Score. Figure S4: Comparison of validation points. red: more than two people think it is non-crop; blue: more than two people think it is crop; green: only one person thinks it is crop. Table S1: Comparison of the accuracy of a BULC-U derived crop map for 2000* compared to the base GFSAD 2015 cropland map, including the overall accuracy and User's and Producer's accuracy of the cropland class. The cropland map for 2000, which we made by subtracting the area of cropland gain identified by BULC-U with a 0.03 slope threshold from the 2015 base map, had similar but slightly lower levels of overall, User's, and Producer's accuracy than the base map.

**Author Contributions:** Conceptualization, L.E., P.B., S.X. and M.A.C.; methodology, P.B., S.X., J.A.C. and M.A.C.; validation, P.B., E.B., S.C.C., M.C. and S.X.; formal analysis, S.X.; data curation, P.B., S.X. and M.C.; writing—original draft preparation, P.B., S.X., M.A.C., L.E. and J.A.C.; writing—review and editing, P.B., L.E., S.X., M.A.C., J.A.C. and M.C.; visualization, S.X. and P.B.; supervision and project administration, L.E.; funding acquisition, L.E. and M.A.C. All authors have read and agreed to the published version of the manuscript.

**Funding:** Support was provided by NASA (80NSSC18K0158) and the National Science Foundation (DEB-1924309; SES-1801251; SES-1832393). M.A.C.'s efforts were supported by NSERC's Canada Graduate Scholarship—Doctoral (CGSD2-534128-2019).

**Data Availability Statement:** Data products and figures from this analysis are freely available at https://www.dropbox.com/scl/fo/n3vy830cwu7nd13veguwo/h?dl=0&rlkey=atsmu825dy914 dx79rjrxbsm1, accessed on 17 August 2022. Processing scripts can also be found at https://github.com/agroimpacts/ptacc/tree/master/Script, accessed on 17 August 2022. Any questions or requests for access can be directed to pbaltezar@g.ucla.edu.

**Acknowledgments:** We would like to express our deepest appreciation to folks who helped contribute to this study in a meaningful way. We are greatly indebted to Su Ye and Lei Song for their insight and expertise on the application of a Bayesian data fusion approach during the conceptualization stage. This allowed us to have a deeper understanding of our preferred methods. Lastly, we want to give a special thanks to all the authors who remained committed despite the challenges of the pandemic ensuing during this project. We would like to especially dedicate this work to the loved ones that authors lost during that time at the beginning of the COVID-19 pandemic.

**Conflicts of Interest:** The authors have no conflict of interest to report.

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
