# Peer review of "Probabilistic Tracking of Annual Cropland Changes over Large, Complex Agricultural Landscapes Using Google Earth Engine"

_remotesensing, doi:10.3390/rs14194896_

Round 1
Reviewer 1 Report
1. I suggest that the authors limit the number of keywords in the manuscript to less than 5.
2. I suggest that the authors change the span of research time to 2000-2020.
3.Line 90, the 2 of km2 in "a large country (~750,000 km2)" should be superscripted.
4.I suggest deleting the content of lines 109-116 and moving the content explained in Fig.1 to this location.
5.Line 127, what is SSA?
Author Response
Thank you to reviewer 1 for taking the time to provide constructive feedback on the structure of the manuscript, grammatical improvements, adjusting the flow of content, and the suggestion for extending the time series.
Thank you for also pointing out that the introduction, presentation of results, and conclusion can be improved. We took the time to add more content to the introduction, results, and conclusion. Let us know again if you feel we need additional figures to properly present our results. We will continue discussing this with our co-authors and make suggestions for more additions if need be.
Point 1: I suggest that the authors limit the number of keywords in the manuscript to less than 5.
We agree and reduced the "keywords" to less than 5. Please let us know if you only allow one word delimited by commas. In our keywords, we took the liberty of using multiple words delimited by a comma.
Point 2: I suggest that the authors change the span of research time to 2000-2020.
Thank you kindly for this suggestion. I agree with this revision because it will provide more current insight into the nature of cropland expansion in our study area. Providing a more current analysis would further increase the value of the research applications to agricultural stakeholders in the area who are dealing with food insecurities today, not ~5 years ago.
Although I would like to accept this revision, we must reject it because we do not have enough time to re-run the analysis, produce new results/figures, and incorporate additional writing, all in time for submission in the next week. Although, we are willing to come back to this and accept if we are given more time. We have asked for additional time and hope that we can pull off this revision given more time.
Point 3: Line 90, the 2 of km2 in "a large country (~750,000 km2)" should be superscripted.
Thank you for catching this oversight. We accept this revision and changed "Km2" of line 90 to be superscripted.
Point 4: I suggest deleting the content of lines 109-116 and moving the content explained in Fig.1 to this location.
Thank you for suggesting that we clean up lines 109 to 116 and move the content from Fig.1 to that section. We agree with this suggestion and believe that it will make this section clear and have a better flow for easier reading. We accept this revision and made the relevant changes. In addition to this, we provided a more succinct Fig. 1 description.
Reviewer 2 Report
In this work a probabilistic tracking of annual cropland changes over large, complex agricultural landscapes using Google Earth Engine was proposed.
The analysis carried out by the authors is extremely interesting. However the work would appear to be in some of the section poor, badly described and missing in some of its parts and therefore it can be published at present. A brief literature review was suggested in the introduction section about Landsat, S1 and ALOS satellite missions application in the agricultural context.
Major revisions need to be made before the paper can be published, especially in the structure and results of the paper.
Below are some general and specific comments.
General comment
Abstract: At the end of the abstract section some of the results achieved in the work could be added such as some accuracy in unsupervised classifications.
Introduction: the authors have talked a lot about the Landsat, S1 and ALOS satellite missions, however, there appears to be a lack of a brief literature review highlighting the various applications in the agricultural context for crop recognition with these missions. Also missing appears to be the GAP in the literature that the paper would go to fill. This can actually be picked up at the end of the introduction in the objectives but could be better described.
Workflow: in the workflow probably appears to be missing data pre-processing, I suggest to add this section in the second column if possible (interpolation, filtering, clipping, etc). Also why does the graph in item 7 "shapelet analysis" appear to have the x-axis (time) descending? I recommend turning the figure around and making the profile go from the year 2000 to 2015.
Materials: some materials need more explanation. For example, how is the GFSAD provided? In what format (vector/raster?)? In what reference system? On which portal? Is it free of charge? Spatial resolution?. Please include all the information so that even those who do not know the datum can learn about it. The same thing should be added for the Landsat datum (for ALOS and S1 data it is done but can be improved), such as spatial resolution, temporal, spectral etc... The description of the data is a key step.
Classification and validation: why did you choose to use a 25x22 km grid? Please specify.
In the unsupervsed classification stage, based on what criterion were 20 classes selected? Please justify in the text the choices made.
NDVI time series: at line 283 you talk about NDVI time series but it has not been described previously, please integrate in the previous sections these parts.
Finally, one last general comment is with respect to acronyms. I recommend reviewing the entire text to check that all acronyms are stated and avoid repetition. it also allows you to check that the acronym is mentioned upstream and not downstream in the paper.
Specific comment
Line 71: definire LULC
Line 88: Use a more impersonal form, limiting the use of words such as "we"
Line 90: check km2
Line 101: you mean “Google earth engine”?
Line 120: define GFSAD
Line 121: define Bulc-U
Line 127: define SSA
Line 130-136: In this section you state that 3 regions can be found in your area based on certain conditions. Please add after this statement a bulleted list in which you clearly define the three sections (I, II, III)
Figure 1: The caption lacks the reference system used to represent the figure.
Table 1: Is not mentioned in the text. My advice is to include it after explaining the Landsat, S1 and ALOS data.
Figure 2: validation points do not appear in the map
Line 215: check “classification First”.
Line 237: are the years of analysis 15 or 16? In case do a check in the whole document
Figure 4: check la caption, Figure 4b is not mentioned.
The following literature could be integrated to your work to improve scientific basis
Line 38-40: reference work about potential of remote sensing in Africa “Mutanga, O., Dube, T., & Galal, O. (2017). Remote sensing of crop health for food security in Africa: Potentials and constraints. Remote Sensing Applications: Society and Environment, 8, 231-239.”
Line 44: Recent work on Congo (Africa) on land management with Earth Observation data. “De Marinis, P., De Petris, S., Sarvia, F., Manfron, G., Momo, E. J., Orusa, T., ... & Borgogno, E. M. (2021). Supporting Pro-Poor Reforms of Agricultural Systems in Eastern DRC (Africa) with Remotely Sensed Data: A Possible Contribution of Spatial Entropy to interpret Land Management Practices. Land, 10(12), 1368.”
Line 272-274 Similar trends reported in the following recent work “Sarvia, F., De Petris, S., & Borgogno-Mondino, E. (2022). Mapping ecological focus areas within the EU CAP controls framework by Copernicus Sentinel-2 data. Agronomy, 12(2), 406”
line 283-290: Work on the use of landsat in the establishment of temporal profiles of NDVI “Hilker, T., Wulder, M. A., Coops, N. C., Seitz, N., White, J. C., Gao, F., ... & Stenhouse, G. (2009). Generation of dense time series synthetic Landsat data through data blending with MODIS using a spatial and temporal adaptive reflectance fusion model. Remote Sensing of Environment, 113(9), 1988-1999.”
Author Response
We are especially thankful for your insightful and highly detailed review of our paper as we know this level of feedback is not easy. We are grateful for the time you took to be helpful, careful, and critical about your review. We agreed with your assessment that this manuscript needed major revisions.
Some of your really helpful comments include expanding on the basis of our methods given other papers that do similar work with NDVI time series analysis, background on the sensors, better pre-processing, and accuracy assessments. We truly feel that these recommendations pushed us to refine a more refined remote sensing focused paper. Because of your review we also feel our figures are much more clean and effective.
We have done our best to fully implement your suggestions. Although we could not fully integrate the entirety of the suggestions given, we hope that you find our integration of your review to be substantial and effective. We look forward to your time once more as you go over our responses. We also want to say we are very thankful considering how lengthy this process has been and appreciate your understanding. Our team experienced some extenuating medical circumstances and conflicting commitments. Thanks again for your valuable time. Attached is also our full cover letter that covers all revisions.
Sincerely,
Priscilla
Response to Reviewer 2
Point 1 Abstract: At the end of the abstract section, some of the results achieved in the work could be added such as some accuracy in unsupervised classifications.
Thank you for pointing out the missing parts in the abstract, and we have made several changes. We added a quantitative description of the accuracy of cropland expansion maps and agree that this would make for a stronger abstract. You can find these changes in the abstract starting on line 27. In addition to this, we saw a need to also expand on our accuracy assessments. In addition to the accuracy assessments (see table 3 as well, new figure added), we conducted an additional evaluation of the unsupervised cluster using a ‘purity test’. We examined if the unsupervised cluster falls within a single landcover type when overlapping with a landcover map. With the cluster purity test we calculated the proportion of each segment that was occupied by the dominant k-means cluster falling within it. We think this purity test helps to understand the quality of the unsupervised classification. However, we are concerned about including this test in the paper, as it may distract from our primary focus on the BULC-U analysis and will therefore keep these additional accuracy assessments in our supplementary information. BULC-U was designed to continually update a time series of unsupervised classifications given the provisional probability of a pixel being a certain class. If you think the unsupervised classification is an essential component of this paper, we will include it in the main text or supplement material.
Point 2 Introduction: The authors have talked a lot about the Landsat, S1, and ALOS satellite missions, however, there appears to be a lack of a brief literature review highlighting the various applications in the agricultural context for crop recognition with these missions. Also missing appears to be the GAP in the literature that the paper would go to fill. This can be picked up at the end of the introduction in the objectives but could be better described.
We extended our literature review to address this concern, by including additional examples from the literature on lines 81 – 104 in the revised manuscript. We emphasized in our edits that SAR is effective for agricultural remote sensing due to its unique capability to actively illuminate targets regardless of atmospheric effects or time of day, as well as its sensitivity to the geometric structure and dielectric properties of crops. In addition to that we included additional studies that show the utility of using multiple sensor combinations either purely with different radar sensors or across radar and optical sensors. Most importantly one of the case studies we added indicated that in order to enable more operation time series analysis in areas with cloud cover, which is the case for our study area, it highlighted the importance of including radar. We hope that these changes are sufficient and can give more background.
Point 3 Workflow: The workflow probably appears to be missing data pre-processing, I suggest adding this section in the second column if possible (interpolation, filtering, clipping, etc.). Also, why does the graph in item 7 "shapelet analysis" appear to have the x-axis (time) descending? I recommend turning the figure around and making the profile go from the year 2000 to 2015.
We re-designed the figure to include more details in our workflow. We moved the data preprocessing to the first column. We added the intermediate steps for preparing BULC-U analysis in the second column. We changed the profile to go from the year 2000 to 2015. You can find our new figure for our methods in the revised manuscript.
Point 4 Materials: Some materials need more explanation. For example, how is the GFSAD provided? In what format (vector/raster?)? In what reference system? On which portal? Is it free of charge? Spatial resolution? Please include all the information so that even those who do not know the datum can learn about it. The same thing should be added for the Landsat datum (for ALOS and S1 data it is done but can be improved), such as spatial resolution, temporal, spectral, etc... The description of the data is a key step.
The GFSAD dataset is described in section 2.2.1 of the basemap. It is a 30 m dataset called the Global Food Security-Support Analysis Data (GFSAD) cropland dataset. We agree with the suggestion to add more details on this specific dataset. We also agree we are missing background on the other sensors in our study. We have made additional changes which can be seen on lines 189 - 201. We included additional papers to our references from each sensor’s respective product manual provided through each sensor agency. We thank the reviewer for pointing out this critical step in our writing.
Point 5 Classification and Validation: Why did you choose to use a 25x22 km grid? Please specify. In the unsupervised classification stage, based on what criterion were 20 classes selected? Please justify in the text the choices made.
We added more explanation on lines 257 - 264 about the design of the 25x22 km grid and why 20 classes were selected. In general, the grid size is an arbitrary setting for the ease of processing in GEE. We compared the BULCU crop probability map using 20 classes and alternatives such as 30 and 40 classes. We visually examined the resulting crop probability maps and found minimal differences and thus chose to proceed with a more parsimonious map with fewer classes. The grid size and number of classes can be selected by the end-user, and we encourage the reader to select these settings based on their experiment needs (final accuracy, computing power, etc.)
Point 6 NDVI time series: In line 283 you talk about the NDVI time series but it has not been described previously, please integrate into the previous sections these parts.
We took this comment to refer to the first paragraph of section 2.3.3 Shapelet analysis on lines 338 - 391. We added additional clarification that the NDVI time series is specifically referring to literature that used the shapelet algorithm specifically for a time series of NDVI, which is not related to what we did in our study. We added on line 353 that we adapted the shapelet to detect cropland gain for a time series of cropland probabilities. We hope that this additional text makes for a clear distinction. We also feel that because we are only talking about the NDVI time series regarding the referred study, we do not need extra information. Readers can read the referred studies in order to understand how they used the NDVI time series. It is assumed that the reader will understand that our study specifically looks at crop probabilities as is described in lines 353 - 354.
Point 7 Acronyms: I recommend reviewing the entire text to check that all acronyms are stated and avoid repetition. it also allows you to check that the acronym is mentioned upstream and not downstream in the paper.
We have reviewed the definition and uses of our acronyms, including Google Earth Engine (GEE), Normalized Difference Vegetation Index (NDVI), Green Chlorophyll Vegetation Index (GCVI), and Global Food Security-Support Analysis Data (GFSAD). We went through entire document to make sure that the full term with appropriate spelling is mentioned first followed by the acronym for entirety of document.
Point 8 Specific Comments:
Line 71: define LULC
Changed.
Line 88: Use a more impersonal form, limiting the use of words such as "we"
Thank you for this suggestion, and we understand that the more personal form has traditionally been considered inappropriate. We went forward with changing the text throughout to not use this conjugation. That being said, there are certain areas of the text where some use of the impersonal was used to maintain more succinct text.
Line 90: check km2
Revised
Line 101: you mean “Google earth engine”?
Revised
Line 120: define GFSAD
Revised
Line 121: define Bulc-U
Revised
Line 127: define SSA
Revised
Line 130-136: In this section you state that 3 regions can be found in your area based on certain conditions. Please add after this statement a bulleted list in which you clearly define the three sections (I, II, III)
Thank you for pointing out that this may not be clear. In actuality, the agroecological regions are called that; region I, region II, and region III. We capitalized regions on line 165 to make it more apparent that this is the official agroecological name in addition to adding that it is the countries agronomist classification of the regions. We hope that these minor changes make it clearer. We also did not include the in-detail information of each region because that is outside of the focus of our paper. Thank you for pointing it out though.
Figure 1: The caption lacks the reference system used to represent the figure.
Thank you for pointing this out, but we believe our caption of figure 1 does not necessarily need the reference system information. It would be unclear to add that level of information to a caption that is meant to capture our entire workflow and is including multiple different datasets in different processing environments. We also want to note that according to the GEE recommendations for handling data because with scale, the projection in which computations take place is determined on a pull basis, that is input are requested in the output projection. Once we pulled processed data from GEE to python or other GIS, we used the xxxxx reference system. Please refere to section xxx on lines xxx to see our description of projections and reference systems.
Table 1: Is not mentioned in the text. My advice is to include it after explaining the Landsat, S1 and ALOS data.
We agree with this comment and we moved table 1 till after the description of satellite images used starting on line 250.
Figure 2: validation points do not appear in the map
We have edited our study area figure found on line 184.
Line 215: check “classification First”.
Revised
Line 237: are the years of analysis 15 or 16? In case do a check in the whole document
2000-2015, should be 16 years. We changed all “15 years” to “16 years”.
Figure 4: check la caption, Figure 4b is not mentioned.
Revised
Point 9 Integrating additional literature
Line 38-40: reference work about the potential of remote sensing in Africa “Mutanga, O., Dube, T., & Galal, O. (2017). Remote sensing of crop health for food security in Africa: Potentials and constraints. Remote Sensing Applications: Society and Environment, 8, 231-239.
Thank you for suggesting additional literature that would better support the content of this manuscript. Similarly, we felt that there would be easier integration of this paper in another section in line 74 as an additional citation as well.
Reviewer 3 Report
Abstract: Generally, methods are in the past tense, please reconsider: “build” should be built, “map” should be “mapped”. Overall, the abstract may seem unreasonable, if the results are similar to other studies. Could have been focused on the utility or novelty of this study, which is given later.
Authors may consider adding a brief section on Study Area for the global readers; how is agriculture in Zambia, its location, climate, major crops, etc.
It should be easy to extend the study period to 2021, especially when using GEE and the authors' claim about the robustness of the proposed methodological framework.
I found the study area section 2.1. Should be moved up Before the Methods or any other reasonable place.
Is this a continuous assessment (annual from 2001 to 2015) or bi-temporal 2001 and 2015?
Line 171: Which bands exactly, seems it is all that Landsat images have except Thermal bands.
172: Add the abbreviations for the indices.
Calculating the medoid for the annual composites might be risky. We don’t know about the number of available images each year and their temporal coverage. This might give inconsistent results. Generally, such studies define a specific temporal window covering the growing season(s) or time of year with maximum, possible, available images
Figure 5: Why the SLC-off data was used for 2015, when OLI was available, similarly, for previous years TM data could have been given priority over the ETM+. To avoid the SLC-off missing data.
Author Response
We are especially thankful for your insightful review of our paper as we know this level of feedback is not easy. We are grateful for the time you took to be helpful, careful, and critical about your review. We agreed with your assessment that this manuscript needed major revisions.
We absolutely agree that this type of work merits an extension of this work to 2021. We do believe that because of the robustness of this work that it is not only feasible, but already in our commitments to do so in our upcoming papers. But, out of respect of time and effort of reviewers and rolling this out for the special issue, the additional changes to figures, writing, results interpretation warrants that we focus those efforts for the next paper. We will also carefully reconsider the use of different compositing methods in order to not introduce possibly inconsistent results.
We have done our best to fully implement your suggestions. Although we could not fully integrate the entirety of the suggestions given, we hope that you find our integration of your review to be substantial and effective. We look forward to your time once more as you go over our responses. We also want to say we are very thankful considering how lengthy this process has been and appreciate your understanding. Our team experienced some extenuating medical circumstances and conflicting commitments. Thanks again for your valuable time.
Sincerely,
Priscilla
Response to Reviewer 3
Point 1 Abstract: Generally, methods are in the past tense, please reconsider: “build” should be built, “map” should be “mapped”. Overall, the abstract may seem unreasonable, if the results are like other studies. Could have been focused on the utility or novelty of this study, which is given later.
The Abstract was revised to address this comment.
Point 2: Authors may consider adding a brief section on Study Area for the global readers; how is agriculture in Zambia, its location, climate, major crops, etc.
We agreed that we could add additional background on the nature of Zambia’s agriculture in terms of where we are seeing growth, the way that it is changing, and the main crops of the country. We did also add additional climate characteristics of our study area in Zambia. We hope that we added enough detail on lines 161 – 185 to suffice this suggestion.
Point 3: It should be easy to extend the study period to 2021, especially when using GEE and the authors' claim about the robustness of the proposed methodological framework.
Thank you, Reviewer 1 made a similar point, and we agree that it would be valuable to extend the analysis. However, as we wrote in response to Reviewer 1, we feel that this update is beyond the scope of this paper, which is focused on developing and demonstrating the methodology, and we plan to do that longer assessment as part of subsequent work focused on analyzing the dynamics of cropland change. An excerpt of our response to Reviewer 1 is pasted below, for further clarification on this point:
We in fact plan to make such an update in future work, which will focus on analyzing the nature of changes in the past two decades. It will also require substantial development effort to the data inputs, such as a more accurate and recent base map for ~2020. Given that additional effort, and the fact that this particular paper is focused primarily on demonstrating the overall methodology, we feel that making the update is beyond the scope of this paper.
Point 4: I found the study area section 2.1. Should be moved up Before the Methods or any other reasonable place.
We definitely agree with this comment and moved the entire section 2.1 to line 143. It makes more sense to set the scene of our study area’s geography first before the description of the datasets used for analysis.
Point 5: Is this a continuous assessment (annual from 2001 to 2015) or bi-temporal 2001 and 2015?
The assessment is bi-temporal. We only created validation samples for 2000, 2010, and 2015. Therefore, the samples from three years allow us to do an assessment using three pairs of years; (1) 2000 and 2010; (2) 2010 and 2015, and (3) 2000 to 2015. We used 2000 to 2015 as our primary assessment for BULC-U for detecting cropland expansion. We also used 2000 and 2010, and 2010 and 2015 to assess whether shapelet analysis got the correct cropland expansion year.
Point 6 Line 171: Which bands exactly, seems it is all that Landsat images have except Thermal bands.
We made it clear by adding red, green, and blue after “visible”. So yes, for Landsat we used RGB, NIR and SWIR.
Point 7: 172: Add the abbreviations for the indices.
Added
Point 8: Calculating the medoid for the annual composites might be risky. We don’t know about the number of available images each year and their temporal coverage. This might give inconsistent results. Generally, such studies define a specific temporal window covering the growing season(s) or time of year with maximum, possible, available images
Yes. We tried using both composites, i.e., the growing season plus off-season only and the annual as our temporal window. Visually result looks similar. However, using growing seasons and off seasons will lead to more pixels as no observations and impact the spatial completeness of the data. So eventually we used the annual composite.
Point 9 Figure 5: Why the SLC-off data was used for 2015, when OLI was available, similarly, for previous years TM data could have been given priority over the ETM+. To avoid the SLC-off missing data.
According to many studies, the Landsat 8 OLI data was not consistent with the Landsat 5 TM and Landsat 7 ETM+. For example, OLI-derived NDVI is usually higher than TM and ETM+ derived. NDVI. So we avoided using Landsat 8 OLI to keep the consistency in the observation, despite the Landsat 7 SLC-off issue. We also found Landsat 5 TM had more missing data than Landsat 7 ETM+ in Zambia in the early years (2000-2010). Therefore, eventually, we decided to use Land 7 ETM+ including the SLC-off data.

Round 2
Reviewer 2 Report
The authors have responded to all the comments made, the work now seems to be more robust and clear. However, before proceeding with publication, I would still advise the authors to carry out a further check of the manuscript for a revision of the English and to check some typing errors, e.g. figure 2 is presented before figure 1. Once these revisions are completed, I think the work is publishable.
Author Response
Thank you for taking the time once more to go over our edits from Round 1. You pushed us to refine this manuscript to be more robust in it's remote sensing approach with a lot of careful detail. Here are our responses to the minor revisions for this round
- Point 1: Fix the incorrectly named figures and the relevant text. Proofread the manuscript for any other remaining errors.
- Thank you so much for pointing this out! This error was addressed by changing the naming of the figures on lines 151, 172, and 177.
- We are definitely happy you asked us to do another proofread, and we found one more error on lines 91 - 94 where we talk about an additional radar study that also uses multiple sensors for the sake of increasing observations.
- we changed the syntax of lines 204-205 for easier reading
- line 229 add "been"
- line 262 replace w/"observations"
- line 404 replace w/"imagery"
- line 543 add "that were"
- line 568 delete the comma
- line 567 delete extra space
- line 606 add "showing the", add "showing"
- line 646 delete "than would be "
- line 647 delete extra spaces
- line 658, delete "in order"
- line 842 delete spaces
- line 865 delete the extra "References" text